# Non-myopic Generation of Language Models for Reasoning and Planning

**Chang Ma**[♠]   **Haiteng Zhao**[♡]     **Junlei Zhang**[◇△]   **Junxian He**[*♣]   **Lingpeng Kong**[*♠]
[♠]The University of Hong Kong   [♡]Peking University   [◇]Zhejiang University
[△]Westlake University   [♣]The Hong Kong University of Science and Technology
{cma, lpk}@cs.hku.hk, junxianh@cse.ust.hk

## Abstract

Large Language Models (LLMs) have demonstrated remarkable abilities in reasoning and planning. Despite their success in various domains, such as mathematical problem-solving and coding, LLMs face challenges in ensuring reliable and optimal planning due to the inherent myopic nature of autoregressive decoding. This paper revisits LLM reasoning from an optimal control perspective, proposing a novel method, Predictive-Decoding, that leverages Model Predictive Control to enhance planning accuracy. By reweighting LLM distributions based on foresight trajectories, Predictive-Decoding aims to mitigate early errors and promote non-myopic planning. Our experiments show significant improvements across a wide range of tasks in math, coding, and agent-based scenarios. Furthermore, Predictive-Decoding demonstrates computational efficiency, outperforming search baselines while utilizing inference compute more effectively. This study provides insights into optimizing LLM planning capabilities. Code is available at this repo.

## 1 Introduction

Large Language Models (LLMs) are extensively pretrained on large corpus to predict the next tokens. Models like GPT-4 (Achiam et al., 2023) have demonstrated a capacity for step-by-step reasoning and planning, breaking down complex problems into sequential steps that progressively lead to solutions (Wei et al., 2022). This *sequential planning* capability has led to significant advancements in mathematical problem-solving (Cobbe et al., 2021) and programming tasks (Chen et al., 2021). It has also enabled essential applications such as tool use (Qin et al., 2023) and the development of LLM agents (Yao et al., 2022). Consider a scenario where an agent is tasked with cooking a meal: given an instruction like "*Put salt on steak*" and an initial observation such as "*There is a shelf in front of you*," the LLM assigns the highest probability to the most appropriate action. After executing this action, the LLM generates subsequent actions, each building upon the previous steps. This process creates a sequential chain of actions that advances toward task completion, guided by continuous interaction with environmental observations:

$$p(\text{ ``Go to the shelf.''} \mid \text{``There is a shelf in front of you.''})$$
$$p(\text{ ``Pick up the saltbottle.''} \mid \text{``Go to the shelf. A saltbottle is on the shelf.''}),$$

where "*A saltbottle is on the shelf*" is the new observation after going to the shelf. This iterative process continues until the specified goal state is achieved. However, this autoregressive decoding process could easily lead to irreversible errors, as LLMs tend to follow the most natural local follow-ups. For instance, given a different observation "*A spice bottle is on the shelf*", the LLM is likely to generate actions "*Pick up the spice bottle*" followed by "*Shake the bottle over the steak*". Such a sequence of actions totally conflicts with the intended instruction "*Put salt on the steak*". This contrasts with reinforcement learning paradigms, which use reward-based training to teach models to anticipate long-term action consequences and learn optimal decision policies (Silver et al., 2014). The short-sightedness of LLMs raises critical questions about their planning capabilities: (i) Can an LLM proactively avoid erroneous steps without necessitating their occurrence? (ii) What is the degree of optimality achievable in LLM-based planning strategies?

---

[*]Equal advising.

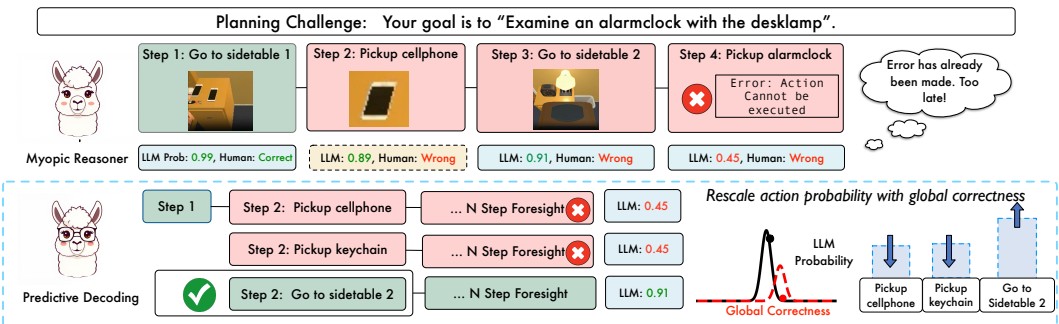

Figure 1: The illustrative overview of Predictive-Decoding on one ALFWorld example. LLM autoregressive planning often suffers from near sight. Predictive-Decoding rescales LLM generation distribution based on evaluation of foresight, enabling non-myopic planning.

To investigate the causes and extent of myopia (short-sightedness) in LLM planning, we refer to optimal control theory (Qin & Badgwell, 1997). Optimal planning is framed as a combinatorial optimization problem aimed at maximizing goal achievement through $T$ steps of actions $(a_1, a_2, \ldots, a_T)$. Model Predictive Control (MPC; Camacho et al., 2007) outlines the optimality condition for LLM-like planners – sequential planning must be non-myopic, which requires the anticipation of future outcomes in addition to considering past actions, represented as $p(a_t \mid History, Future)$. Conversely, focusing solely on $p(a_t \mid History)$ can lead to irreversible mistakes and potential planning failures. Recent studies (Shih et al., 2023; Bachmann & Nagarajan, 2024) have observed myopia in language models pretrained with next-token prediction. Our research further reveals that even in advanced LLMs like `Llama-3` (Dubey et al., 2024), more than half of the reasoning processes on popular math tasks demonstrate a lack of global awareness, with myopia being particularly pronounced in incorrect instances (§3.1). Additionally, LLMs initially do not seem sensitive to mistakes; however, errors accumulate during multi-step planning and become increasingly noticeable over several steps (§3.2). These findings emphasize the importance of thinking with foresight.

To address this, we propose Predictive-Decoding, a training-free approach to improve LLM planning with non-myopic generation. Predictive-Decoding re-weights LLM distribution based on the closed-form solution in Model Predictive Control (Eq. 2), such that sampling from the re-weighted solution is approximately solving planning with global-optimality. For each step $a_t$, Predictive-Decoding samples multiple foresight trajectories $(a_t, a_{t+1}, \ldots, a_{t+T_0})$ and rescales the original generation distribution based on the evaluations of foresight trajectories. Best action $a_t$ is sampled from this distribution. An overview is illustrated in Figure 1. Extensive experiments on math, coding, and agent tasks show notable improvements. For example, without using any additional supervision, we achieve a 7.2% improvement on GSM8K (Cobbe et al., 2021) and a 25.3% improvement on AlfWorld (Shridhar et al., 2021) accuracy over ReAct (Yao et al., 2022). It requires no special prompt design and preserves generation diversity.

Furthermore, Predictive-Decoding utilizes inference compute more effectively than other planning approaches. Previous methods have implemented various techniques to enhance planning, such as repeated sampling (Wang et al., 2022) and searching algorithms (Yao et al., 2024; Xie et al., 2024; Hao et al., 2023; Zhou et al., 2023). While these methods can improve performance, they often require significantly higher computational resources, resulting in sub-optimal computation-performance trade-offs (Snell et al., 2024; Wu et al., 2024b). In contrast, Predictive-Decoding adopts a "no-regrets" approach that optimizes sequentially and simplifies the solution space, avoiding the exponential growth associated with iterative planning methods. This efficiency is reflected in our inference scaling law results with a fixed LLM and reward model, where Predictive-Decoding achieves Pareto superiority over all state-of-the-art (SOTA) baselines (§5.4). Specifically, on the GSM8K benchmark, Predictive-Decoding outperforms MCTS (Hao et al., 2023) by 2.4% and Guided Decoding (Xie et al., 2024) by 1.4% , while utilizing 50% of the FLOPS, demonstrating its effectiveness and efficiency.

## 2 PROBLEM FORMULATION: AN OPTIMAL CONTROL VIEW

One of the fundamental problems in decision-making is whether we can solve a long-term goal step by step (Qin & Badgwell, 1997). Traditionally, reinforcement learning approaches have relied on

Table 1: Non-iterative planning is efficient and suitable for closed-loop settings. Iterative planning, though better for global optimal decisions, is slow and impractical for realistic agent tasks. Predictive-Decoding combines the best of both. Advantages include: **G**lobal-Optimality of solution, **S**caling to long-range planning, **E**fficiency, and **C**losed-Loop Interaction.

| Decision Making | Paradigm Illustration | Representative Methods | Type | Advantage |
|---|---|---|---|---|
| **Iterative Planning** | | Self-Consistency (Wang et al., 2022) | Sample | |
| | | Tree-of-Thought (Yao et al., 2024) | Search | |
| | | Guided-Decoding (Xie et al., 2024) | Search | G |
| | | Monte Carlo Tree Search (Hao et al., 2023) | Search | |
| | | Reflexion (Shinn et al., 2024) | Prompt | |
| **Sequential Planning** | | Chain-of-Thought (Wei et al., 2022) | Sample | S E C |
| | | ReAct (Yao et al., 2022) | Sample | |
| | | Predictive-Decoding (Ours) | Sample | G S E C |

multiple iterations of training (Silver et al., 2014) and searching (Silver et al., 2017) to achieve optimal results. However, recent advancements demonstrate that LLMs possess the capability for sequential planning (Wei et al., 2022). Sequential planning can achieve optimal outcomes if non-myopic, while also offering other benefits, as shown in Table 1. In this section, we first outline the context of the problem and reconsider LLM planning from the perspective of trajectory optimization.

**Planning as Trajectory Optimization** We consider a Decision Process defined by tuple $\langle g, \mathcal{S}, \mathcal{A}, \mathcal{T}, \mathcal{R} \rangle$, with goal $g$, state space $\mathcal{S}$, valid actions space $\mathcal{A}$, transition function $\mathcal{T} : \mathcal{S} \times \mathcal{A} \to \mathcal{S}$, and $\mathcal{R} : r(\mathbf{s}, \mathbf{a}|g) \mapsto \mathbb{R}^+$ is the reward function that measures how well each step aligns with the goal (with the goal dependency omitted for simplicity). *Global-Optimal Planning* aims to find an action sequence that maximizes cumulative reward over a set number of steps. We model this as a Trajectory Optimization process: find a sequence of actions $\mathbf{a}_{0:T}$ that maximizes an objective $\mathcal{J}$ (Return) factorized over per-timestep rewards $r(s_t, a_t)$ over a planning horizon with $T$ steps:

$$\max_{\mathbf{a}_{0:T}} \mathcal{J}(\mathbf{s}_0, \mathbf{a}_{0:T}) = \sum_{t=0}^{T} r(\mathbf{s}_t, a_t),$$

$$\text{subject to } \mathbf{s}_{t+1} = \mathcal{T}(\mathbf{s}_t, a_t), \quad \forall t \in [0, T-1] \tag{1}$$

Conversely, a *Myopic Planner* maximizes the return on a shorter horizon. For instance, a planner that maximizes the immediate reward at current step, i.e $a_t = \max_{a_t} r(\mathbf{s}_t, a_t)$, is myopic, while *non-myopic planning* works towards global-optimality.

Previous approaches to trajectory optimization follow a straightforward intuition: using large language models (LLMs) to generate diverse trajectories through sampling or search, and then selecting the best one (Wang et al., 2022; Yao et al., 2024; Hao et al., 2023; Xie et al., 2024; Shinn et al., 2024). However, a significant drawback is the exponential growth of the solution space. Each action step offers $|\mathcal{A}|$ possibilities, leading to $|\mathcal{A}|^T$ potential solutions after $T$ steps. Despite attempts to prune trajectories and accelerate search (Zhuang et al., 2023; Wang et al., 2024a), most iterative planning methods are either time-consuming or perform poorly with limited iterations (Chen et al., 2024b). Additionally, iterative methods are often unsuitable for closed-loop scenarios, where agents interact with their environment in real-time, as each decision alters the environment and is irreversible.

**Sequential Planning with Model Predictive Control** Model Predictive Control (Qin & Badgwell, 1997) introduces a different paradigm for planning: instead of optimizing an entire action sequence $\mathbf{a}'_{0:T}$ at once, this method selects the best action at each timestep $a'_0, \ldots, a'_t$, fixing each as it progresses and then optimizing the subsequent steps. This transforms Eq. 1 into a series of optimization problems, each with linear size solution space $|\mathcal{A}|$. The colors denote: the current action to decide $a_t$, *history*, and *future*.

$$\begin{cases} a'_0 = \arg\max_{a_0} \left[ \underbrace{\max_{a_1, \ldots, a_T} \mathcal{J}(\mathbf{s}^0; a_0, a_1, \ldots, a_T)}_{\text{Tail Subproblem}} \right] \\ a'_1 = \arg\max_{a_1} \left[ \max_{a_2, \ldots, a_T} \mathcal{J}(\mathbf{s}^0, a'_0; a_1, \ldots, a_T) \right] \\ \ldots \\ a'_T = \arg\max_{a_T} \left[ \mathcal{J}(\mathbf{s}^0, \mathbf{a}'_{0:T-1}; a_T) \right] \end{cases} \tag{2}$$

This method requires two steps: first, for each action $a_t$, compute the maximum future outcomes (tail subproblem), and then select the action that yields the best future outcomes. Model Predictive Control can also use sampling for inference, but it improves sampling efficiency in two folds: the solution space for each equation is linear and the size of the tail subproblem is reduced after each step. We further provide sample efficiency theoretical analysis and simulations in Appendix B.1.

**LLM Autoregressive Planning**    Solving Eq. 2 is challenging – We need to identify an action that achieves the best outcome after $T$ steps. LLM provides a straightforward solution to this problem: it leverages its extensive knowledge to anticipate the next best action, $a_t'$, for achieving the goal. This method can be formalized as:

$$
\begin{cases}
a_0' = \underset{a_0}{\arg\max} P^{\text{LLM}}\left(a_0 \mid \mathbf{s}_0\right) \\
a_1' = \underset{a_1}{\arg\max} P^{\text{LLM}}\left(a_1 \mid \mathbf{s}_0, a_0'\right) \\
\dots \\
a_T' = \underset{a_T}{\arg\max} P^{\text{LLM}}\left(a_T \mid \mathbf{s}_0, \mathbf{a}_{0:T-1}'\right)
\end{cases}
\tag{3}
$$

This approach is widely adopted in LLM reasoning and planning, encompassing Chain-of-Thought (CoT; Wei et al. 2022), ReAct (Yao et al., 2022), and Voyager (Wang et al., 2023a). Research has demonstrated that with carefully designed prompts and in-context examples, LLM autoregressive planning can achieve competitive performance in various tasks (Cobbe et al., 2021; Chen et al., 2021; Shridhar et al., 2021). We further investigate whether LLM autoregressive planning is global-optimal.

## 3    DIAGNOSING THE DEFICIENCY OF LLM PLANNERS

Comparing Eq.3 with Eq.2, we observe that LLM-based action selection at time step $t$ differs from the optimal control approach. Instead of maximizing future outcomes in the tail subproblem, LLMs select actions based on the immediate conditional probability $\arg\max_{a_t} P^{\text{LLM}}(a_t \mid \mathbf{s}^0, a_{0:t-1}')$.[1] Consequently, the optimality of LLM planning hinges on two hypotheses: (1) *Non-myopic*: LLMs inherently plan ahead, with current step probabilities accounting for the success of future steps. (2) *Evaluation capability*: The probability distribution in LLMs can effectively substitute for $\mathcal{J}$, distinguishing successful trajectories from failed ones. In the following sections, we will examine and investigate the validity of these two hypotheses.

### 3.1    FINDING 1: LLM AUTOREGRESSIVE PLANNING IS MYOPIC

We begin by investigating whether LLMs inherently plan ahead. While next-token prediction is typically greedy, evidence suggests extensive pretraining might enable LLMs to implicitly plan future tokens (Wu et al., 2024a). We therefore intends to measure whether current LLMs have such ability. We define a metric "myopic gap" to probe whether LLM could plan ahead: a myopic LLM planner would fail to find the optimal trajectory within its distribution.

**Definition 3.1.** *(Myopic Gap for LLM Planning)* Given a language model with distribution $P(a_0, a_1, \ldots, a_T)$, let $\mathcal{P}$ be the support set of the distribution. $\mathbf{a}_{0:T}'$ are generated autoregressively following Eq. 3. Then the myopic gap for planning is:

$$
p^* = \max_{\mathbf{a}_{0:T} \in \mathcal{P}} P(a_0, a_1, \ldots, a_T) - P(a_0', a_1', \ldots, a_T')
\tag{4}
$$

$p^*$ represents the accumulated difference between $P^{\text{LLM}}(a_t \mid \textit{history}, \textit{future})$ and $P^{\text{LLM}}(a_t \mid \textit{history})$ at all steps. A positive gap ($p^* > 0$) indicates that the LLM's action selection is myopic for at least one intermediate step, while $p^* = 0$ proves consistent global-awareness during planning. Here we include a small margin ($p^* > 0.01 \Rightarrow$ myopic) to discriminate myopic and non-myopic cases.

We empirically calculate the myopic gap of `Llama3-8B` on two challenging reasoning tasks, GSM8K (Cobbe et al., 2021) and MATH (Hendrycks et al., 2021). We ensemble the generation of token-level Beam Search (Freitag & Al-Onaizan, 2017) and Predictive Decoding (detailed in §4) as

---

[1]Each token within $a_t$ is generated based on the highest current probability; we express this at the action level for brevity.

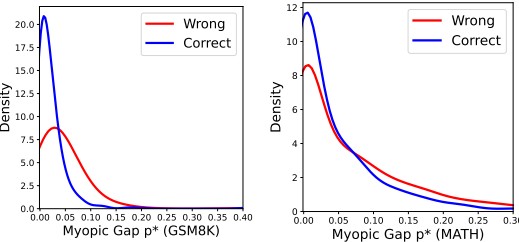

| | Wrong | Correct | Overall |
|---|---|---|---|
| *GSM8K* | | | |
| Myopic | 290 | 539 | 829 |
| Non-myopic | 89 | 401 | 490 |
| Myopic Rate | **76.5%** | 57.3% | 62.8% |
| *MATH* | | | |
| Myopic | 2172 | 785 | 2947 |
| Non-myopic | 1341 | 702 | 2043 |
| Myopic Rate | **61.8%** | 52.7% | 59.1% |

Figure 2: Myopic Gap distribution for correct and wrong samples (drawn with kde-plot). Myopic examples are defined as $p^* > 0.01$. Wrong samples show a higher myopic rate on both tasks.

the non-myopic target. Results in Figure 2 show that around 60% of all examples show are myopic. Additionally, correct samples are often less myopic, while 76.5% of GSM8K errors and 61.8% of MATH errors occur when LLM planning is myopic. Correct samples tend to have smaller gaps, this motivates us that optimizing generation towards non-myopic may lead to improved results.

## 3.2 FINDING 2: LLM STRUGGLES TO IDENTIFY MISTAKES IN PLANNING EARLY

In this section, we explore another bottleneck in LLM sequential planning: Can LLMs evaluate intermediate steps and identify mistakes early on?

We perform an LLM score calibration analysis by comparing LLM evaluations of intermediate steps with ground truth human annotations. We collect samples of trajectories $a_{0:t}, t \in (0, T]$ and use the LLM to evaluate these trajectories $P^{\text{LLM}}(a_0, a_1, \ldots, a_t)$. `GPT-3.5-Turbo` is used to evaluate trajectories on the agent task AlfWorld, while `Llama3-8B` evaluates steps of the GSM8K trajectory. The LLM score for Llama3 is calculated using LLM probability, and the score for proprietary GPT-3.5 is obtained through prompt-based self-evaluation (Xie et al., 2024). Human annotators label each step in the AlfWorld trajectory with scores from $0, 0.25, 0.5, 0.75, 1$, where $0$

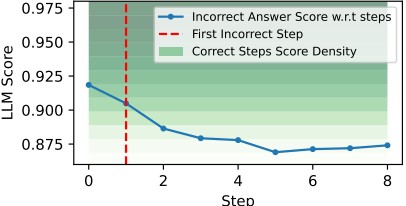

Figure 3: In GSM8K, the first incorrect step's average score is among correct steps, but not after a few steps.

indicates that the task is unlikely to be completed and $1$ indicates a high likelihood of successful completion. Each step in the GSM8K trajectory is labeled as either Correct or Incorrect following Lightman et al. (2023).

As shown in Table 2, `GPT-3.5-Turbo` could barely estimate intermediate progress simply with the current trajectory $\mathbf{a}_{0:t}$, with only $\rho = 0.133$ correlation to human ground truth and Calibration Error 0.165. However, after given future trajectory (foresight of 3 steps), LLM evaluation improves. This is in line with previous observations (Uesato et al., 2022; Lightman et al., 2023) that LLMs are more natural at evaluating complete trajectories, rather than intermediate actions.

The evaluation of intermediate steps for `Llama3-8B` on GSM8K shows a calibration error of 0.332 when compared to human ground truth. We observe that incorporating foresight into the evaluation could help avoid early mistakes. As shown in Figure 3, the score for the first incorrect step falls within the high-density region of correct step scores but becomes easily distinguishable from correct steps after a few iterations. Consequently, the GSM8K calibration score decreases to 0.323 when foresight of three steps is applied (detailed in Table 2). However, the correlation does not improve, as the scores of positive steps slightly drops, leading to some overlap with ambiguous incorrect steps.

Table 2: LLM Score Calibration vs. Human Ground Truth.‡ECE for AlfWorld uses a 0.5 threshold for binary classification.

| Task | Model | Calibration Error ECE (wo. Foresight) | Calibration Error ECE (w. Foresight) ↓ | Correlation $\rho$ (wo. Foresight) | Correlation $\rho$ (w. Foresight) ↑ |
|---|---|---|---|---|---|
| AlfWorld | GPT-3.5-TURBO | 0.165‡ | 0.108‡ ↓ | 0.133 | 0.291 ↑ |
| GSM8K | LLAMA-3 | 0.332 | 0.323 ↓ | 0.417 | 0.376 ↓ |

## 4 BEYOND MYOPIC GENERATION FOR BETTER PLANNING

Our findings presented above highlight critical issues with current LLM decoding methods, particularly their tendency toward myopic planning. In this section, we introduce Predictive-Decoding, which follows MPC principles to reduce myopic planning in LLMs. The pipeline is introduced in following sections and the pseudocode is presented in Algorithm 1.

### 4.1 PREDICTIVE-DECODING

Our decision-making strategy draws inspiration from model predictive control (MPC) (Qin & Badgwell, 1997). It solves the $T$ sub-optimization problems (Eq. 2) and addresses myopia with foresight. Here in order to ensure non-myopic planning, our main objective is to generate $a_t'$ according to:

$$a_t' = \arg\max_{a_t} \mathbb{E}_{\mathbf{a}_{>t}, \mathbf{s}_{>t}} P^{\text{LLM}}\left(a_t, \mathbf{a}_{>t}, \mathbf{s}_{>t} \mid \mathbf{a}_{<t}', \mathbf{s}_{<t}'\right), \forall a_t \in \mathcal{P}, \tag{5}$$

where $\mathcal{P}$ is the support set of distribution $P^{\text{LLM}}(a_t \mid \mathbf{a}_{<t}', \mathbf{s}_{<t}')$, and $\mathbf{a}_{>t} \sim P^{\text{LLM}}(\cdot \mid \mathbf{a}_{<t}', \mathbf{s}_{<t}', a_t)$. Note that based on results from §3.2, LLMs could evaluate future steps after incorporating a few steps of foresight. Therefore we evaluate the future constrained to $T_0$ steps of foresight, i.e. $\mathbf{a}_{>t} := \mathbf{a}_{t+1:t+T_0}$.

We use constrained decoding and sampling to apply soft constraints on outputs, promoting diverse generation and preventing overfitting to our partial ($T_0$ steps) foresight. Our method follows the sampling-importance-resampling (SIR) technique from energy-based models (EBM) (Smith & Gelfand, 1992; Ji et al., 2023) to achieve the optimization goal.

**Proposition 4.1.** *The distribution that solves the optimization problem in Eq.5 is in the form of:*

$$p_\tau(a_t) \propto P^{LLM}\left(a_t \mid \mathbf{a}_{<t}', \mathbf{s}_{<t}'\right) \exp\left[\mathbb{E}_{\mathbf{a}_{>t}, \mathbf{s}_{>t}} P^{LLM}\left(a_t, \mathbf{a}_{>t}, \mathbf{s}_{>t} \mid \mathbf{a}_{<t}', \mathbf{s}_{<t}'\right)/\tau\right]. \tag{6}$$

The detailed proof is in Appendix B.2. Specifically we first use the LLM to sample a set with size K of foresight rollouts $\mathbf{a}_{t:t+T_0}$ given prefix $\mathbf{a}_{<t}'$ in parallel, as well as obtaining the exponential value of generation probability $w_k$ of each rollout. Then we obtain the categorical distribution Categorical $\left(\frac{w_1}{\sum_{k=1}^K w_k}, \cdots, \frac{w_K}{\sum_{k=1}^K w_k}\right)$. The next action step is determined as the first step in the sampled rollout from this distribution. In the limit $K \to \infty$ and temperature $\tau \to 0$, the method recovers the exact maximum value of the distribution. In practice, we can choose $K$ according to computational budget and set $\tau$ according to whether we want the model to fit more towards the distribution or generate more diversely.

Predictive-Decoding mainly uses LLM self-evaluation of foresight to improve planning, but this method could also be integrated with an external heuristics $J$. Eq. 6 can be rewritten as $p(a_t) \propto P^{\text{LLM}}\left(a_t \mid \mathbf{a}_{<t}, \mathbf{s}_{<t}\right) \exp\left[\mathbb{E}_{\mathbf{a}_{>t}} \mathcal{J}\left(\mathbf{s}_0; \mathbf{a}_{0:T}\right)\right]$, which uses the heuristic function $J$ rather than LLM probability to evaluate the foresight. This setting enables LLM to utilize discrete heuristics more effectively and makes it easier to compare with other search baselines which often use external reward models. More details are elaborated in §5.3.

### 4.2 RECYCLING TRAJECTORY ROLLOUTS ♻

In previous work, foresight in generation has been applied for controllable tasks (Deng et al., 2020; Lu et al., 2021; Qin et al., 2022), typically using token-level constraints with a short lookahead of only a few tokens. In contrast, generating future actions involves longer foresights, which could lead to increased computation with complexity $O(KT_0)$. Insufficient sampling number $K$ can inaccurately model the distribution $p_\tau(a_t)$, while larger $K$ is inefficient. Inspired by work on accelerating LLM inference (Fu et al., 2024), we design a memory pool to recycle sampled trajectories.

We observe that different sampled trajectories at different time-steps often overlap. For example, when an agent needs to heat a tomato, the trajectory starting from "*go to the microwave → put the tomato in a microwave → heat the tomato with microwave* " overlaps with "open the fridge → take the tomato from fridge → go to the microwave → put the tomato in a microwave → heat the tomato with microwave" since the tomato needs heating after being taken from the fridge. Thus, previously sampled trajectories and their probabilities $(\mathbf{a}_{t:t+T_0}, p_t)$ can be reused for evaluating timestep $t' \in (t, t+T_0)$, provided their trajectory matches $\mathbf{a}_{t:t'} = \mathbf{a}_{t:t'}'$. Similarly, this could be applied for mathematical reasoning as well as coding, as separate reasoning paths often share steps.

Table 3: The accuracy(%) on GSM8K(eight-shot) and MATH (four-shot).[†] CODEX results are taken from Xie et al. (2024). Inference FLOPS for PAL is denoted as $k$ and Guided-Decoding FLOPS scales according to the number of generated tokens reported in the paper.

| Model | Size | Method | Inference FLOPS | #Sample | GSM8K | MATH |
|---|---|---|---|---|---|---|
| CODEX | 175B | Autoregressive (PAL, Gao et al. (2023))[†] | $k$ | $N = 1$ | 72.0 | - |
| | | Guided-Decoding(Xie et al., 2024)[†] | $138.5k$ | $N = 1$ | 80.2 | - |
| LLAMA-3 | 8B | Autoregressive (PAL) | $7.5 \times 10^{12}$ | $N = 1$ | 71.3 | 29.7 |
| | | Beam Search | $478.6 \times 10^{12}$ | $N = 1$ | 78.4 | **34.4** |
| | | Autoregressive + Self-Consistency | $59.8 \times 10^{12}$ | $N = 8$ | 80.9 | 31.8 |
| | | Beam Search + Self-Consistency | $478.6 \times 10^{12}$ | $N = 8$ | 79.5 | 39.4 |
| | | Guided-Decoding | $119.3 \times 10^{12}$ | $N = 8$ | 63.9 | - |
| | | Predictive-Decoding | $161.3 \times 10^{12}$ | $N = 1$ | **78.5** | 34.0 |
| | | Predictive-Decoding + Self-Consistency | $129.0 \times 10^{13}$ | $N = 8$ | **81.3** | **40.3** |
| MISTRAL-0.3 | 7B | Autoregressive (PAL) | $6.6 \times 10^{12}$ | $N = 1$ | 53.4 | 12.7 |
| | | Beam Search | $418.8 \times 10^{12}$ | $N = 1$ | 65.5 | 18.2 |
| | | Predictive-Decoding | $141.1 \times 10^{12}$ | $N = 1$ | **66.7** | **19.5** |
| LLAMA-3 | 70B | Autoregressive (PAL) | $53.7 \times 10^{12}$ | $N = 1$ | 90.1 | 43.8 |
| | | Beam Search | $343.7 \times 10^{13}$ | $N = 1$ | 91.4 | 48.1 |
| | | Predictive-Decoding | $115.5 \times 10^{13}$ | $N = 1$ | **93.1** | **49.2** |

Table 4: The pass@1(%) and pass@10(%) rates on HumanEval (zero-shot) and MBPP (zero-shot). MBPP uses the 500 samples test set from huggingface. [†]Results are taken from Roziere et al. (2023).

| Model | Size | Method | FLOPS pass@1 | HUMANEVAL | | MBPP | |
|---|---|---|---|---|---|---|---|
| | | | | pass@1 | pass@10 | pass@1 | pass@10 |
| GPT-3.5 TURBO[†] | - | Autoregressive | - | 48.1 | - | 52.2 | - |
| GPT-4[†] | - | Autoregressive | - | 67.0 | - | - | - |
| LLAMA-3 | 8B | Autoregressive | $2.9 \times 10^{12}$ | 44.3 | 72.1 | 33.5 | 48.4 |
| | | Beam Search | $290.0 \times 10^{12}$ | 53.5 | 70.1 | 33.6 | 50.2 |
| | | Self-infilling (Zheng et al., 2023) | $5.8 \times 10^{12}$ | 51.8 | 75.6 | 33.4 | 48.6 |
| | | Predictive-Decoding | $76.6 \times 10^{12}$ | **57.3** | **76.8** | **38.4** | **52.6** |
| DEEPSEEK CODER | 6.7B | Autoregressive | $2.4 \times 10^{12}$ | 45.2 | 66.5 | 38.6 | 53.8 |
| | | Beam Search | $240.0 \times 10^{12}$ | 46.7 | 50.0 | **39.7** | 42.6 |
| | | Self-infilling | $4.8 \times 10^{12}$ | 46.3 | 76.2 | 38.4 | 51.6 |
| | | Predictive-Decoding | $62.4 \times 10^{12}$ | **47.6** | **79.9** | 34.0 | **55.2** |

# 5 EXPERIMENTS

## 5.1 EXPERIMENT SETTINGS

**Benchmarks** Our evaluation covers three domains: *math* - GSM8K (Cobbe et al., 2021) and MATH (Hendrycks et al., 2021), *coding* - HumanEval (Chen et al., 2021) and MBPP (Austin et al., 2021), and *agents* - two agent tasks AlfWorld (Shridhar et al., 2021) and PDDL (from Agentboard, Ma et al., 2024) to understand planning ability in closed-loop interactions.

**Evaluation Settings** We list the LLMs used for evaluation in Appendix F.1. We use vLLM inference infrastructure for efficiency (Kwon et al., 2023). To ensure fair comparisons, we use standardized prompts from Guo et al. (2024), Gao et al. (2023), Cobbe et al. (2021), and Ma et al. (2024). Hyperparameter and prompt details are stated in Appendix F

**FLOPS calculation** Many of our experiments report computational efficiency to better illustrate performance-efficiency tradeoff. We using FLOPS as the metric, Kaplan et al. (2020), i.e. FLOPS $\approx 6nP$, where $P$ is the number of parameters in the LLM, and $n$ the number of generated tokens.

**Baselines** We incoporate iterative planning baselines – Self-Consistency (Wang et al., 2022), Guided-Decoding (Xie et al., 2024), MCTS (Hao et al., 2023) and Self-Infilling (Zheng et al., 2023) – and sequential planning baselines – ReAct (Yao et al., 2022), Beam Search, PAL (Gao et al., 2023), COT (Wei et al., 2022)), Tree of Thought (Yao et al., 2024), A* Search (Zhuang et al., 2023; Wang et al., 2024b) – in our evaluation. We excluded search agents such as LATS (Zhou et al., 2023) from our comparison, as they primarily address non-environment altering agents (e.g., QA and Retrieval).

Table 5: The success rates(%) and the progress rates(%) on AlfWorld (one-shot) and PDDL (one-shot) within 20 steps. Predictive-Decoding uses the same prompt as Act.

| Model | Size | Method | FLOPS | # Sample | ALFWORLD Success | Progress | PDDL Success | Progress |
|---|---|---|---|---|---|---|---|---|
| GPT-3.5 TURBO | - | Act (Yao et al., 2022) | $k$ | $N = 1$ | 6.7 | 25.6 | 5.0 | 20.0 |
| | | ReAct (Yao et al., 2022) | $8.4k$ | $N = 1$ | 7.5 | **34.8** | 3.3 | 15.6 |
| | | Predictive-Decoding | $40.7k$ | $N = 1$ | **14.9** | 33.2 | **11.7** | **28.9** |
| LLAMA-3.1 | 70B | Act | $43.2 \times 10^{12}$ | $N = 1$ | 19.4 | 43.0 | 35.0 | 54.6 |
| | | ReAct | $488.0 \times 10^{12}$ | $N = 1$ | 18.7 | 43.4 | 18.3 | 36.9 |
| | | Predictive-Decoding | $224.1 \times 10^{13}$ | $N = 1$ | **44.0** | **53.9** | **38.3** | **59.7** |

Table 6: Impact of Foresight $T_0$, sampling number $K$ and trajectory recycle ♻. FLOPS of autoregressive generation is denoted as $k$ and is scaled based on number of generated tokens.

| # Foresight $T_0$ | FLOPS | GSM8K | HUMANEVAL | # Sampling $K$ | FLOPS | HUMANEVAL w. ♻, | wo.♻ |
|---|---|---|---|---|---|---|---|
| $T_0 = 1$ | $3.6k$ | 77.0 | 45.1 | $K = 2$ | $6.6k$ | 51.2 | 47.6 |
| $T_0 = 2$ | $7.2k$ | 76.4 | 48.8 | $K = 4$ | $13.2k$ | 54.3 | 46.3 |
| $T_0 = 4$ | $14.4k$ | 77.9 | 52.4 | $K = 6$ | $19.9k$ | 50.6 | 49.4 |
| $T_0 = 6$ | $21.5k$ | 78.5 | 57.3 | $K = 8$ | $26.4k$ | 57.3 | 49.4 |

## 5.2 PREDICTIVE DECODING MAIN RESULTS

**Results on Math Tasks** Following (Xie et al., 2024), we use the Program Aided Language Model (PAL) format, executing generated code in Python to obtain answers. As shown in Table 3, Predictive-Decoding improves accuracy by 7.2% on GSM8K and 4.3% on MATH with Llama3, and 13.3% and 6.8% with Mistral-0.3. While Beam Search achieves similar gains, it requires over 3× the computation. Our method matches or exceeds Beam Search and, when combined with Self-Consistency, outperforms both autoregressive and beam search approaches. Guided-Decoding works well with Codex but underperforms with Llama3 due to overconfident scores. For the stronger 70B model, the myopic gap is narrowed but Predictive-Decoding still improves performances by a distinct margin.

**Results on Coding Tasks** Table 4 presents results for two code generation tasks. The Pass@10 setting selects the best from 10 generations, highlighting both accuracy and diversity. Predictive-Decoding outperforms all baselines with Llama3, achieving a strong quality-diversity balance, as detailed in the subsequent temperature analysis. Interestingly, beam search shows less stability in the Pass@10 setting, suggesting that search-based methods may result in lower diversity in generation.

**Results on Agent Tasks** Table 5 presents the performance of two agent tasks: Predictive-Decoding shows improvements of up to 24.6% on two agent tasks and outperforms consistently over Act and ReAct. Notably, Predictive-Decoding achieved more significant enhancements in success rates than in progress rates. Predictive-Decoding excels in strategic planning and error reduction, leading to higher success rates. Case studies compared to baselines are available in Appendix E.

***Impact* of Foresight Length** $T_0$ We analyze the impact of foresight length in Table 6. Performances on both GSM8K and HumanEval both improve with longer foresight. Notably, `Llama3`'s myopia on GSM8K is mitigated with just 4 foresight steps, indicating that LLMs already possess partial capability to plan ahead. On HumanEval, Predictive-Decoding scale significantly with increased foresight length. This aligns with findings from studies on inference time scaling (Snell et al., 2024), which we discuss further in §5.4. Our method is more efficient in mitigating myopia compared to Beam Search, as shown in Figure 4.

***Impact* of Sampling number** $K$ **and Trajectory Recycle** We must sample from the original distribution as described in §4.1, but more sampling may reduce efficiency. As shown in Table 6, increasing $K$ leads to better performances as it could give a more accurate account of the original generation distribution. Also, trajectory recycle significantly improve the sampling efficiency.

***Impact* of Sampling temperature** $\tau$ **and LLM generation temperature** $\alpha$ As shown in Figure 5 when $\alpha$ remains constant, all values of $\tau$ enhance the diversity compared to vanilla generation, as opposed to search-based methods that often sacrifice diversity. Higher $\tau$ values increase diversity but can degrade performance by causing myopia. $\alpha$ significantly influences performance outcomes, with the best balance between diversity and performance achieved at moderate $\alpha$ values (0.3-0.6).

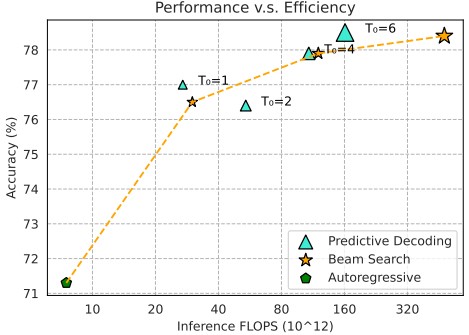
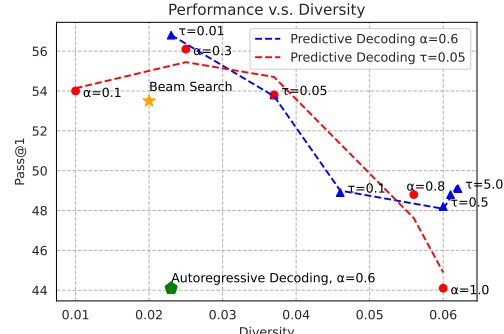

Figure 4: Performance v.s. Efficiency on GSM8K. Predictive Decoding is Pareto superior to Beam Search with longer foresight.

Figure 5: Illustrating Performance and Diversity tradeoff on HumanEval by controlling parameters $\tau$ and $\alpha$. Diversity uses 1- ROUGE score.

Table 7: The success rates(%) and the progress rates(%) on AlfWorld (one-shot) and PDDL (one-shot) with different large language models and matching function as heuristics.

| Model | Method | # Foresight $T_0$ | ALFWORLD | | PDDL | |
|---|---|---|---|---|---|---|
| | | | Success | Progress | Success | Progress |
| GPT-3.5 TURBO | Act (Yao et al., 2022) | $T_0 = 0$ | 6.7 | 25.6 | 5.0 | 20.0 |
| | ReAct (Yao et al., 2022) | $T_0 = 0$ | 7.5 | 34.8 | 3.3 | 15.6 |
| | Predictive-Decoding + Reward | $T_0 = 0$ | 7.5 | 29.5 | 6.7 | 23.9 |
| | Predictive-Decoding + Reward | $T_0 = 6$ | **13.4** | **37.6** | **8.3** | **27.9** |

Table 8: The accuracy(%) on GSM8K (8-shot) with reward model(RM) Math-Shepherd.

| Model | Size | Method | Inference FLOPS | Reward FLOPS | GSM8K |
|---|---|---|---|---|---|
| LLAMA-3 | 8B | Autoregressive | $5.6 \times 10^{12}$ | $0.0 \times 10^{12}$ | 70.4 |
| | | RM-weighted Self-Consistency | $44.5 \times 10^{12}$ | $0.4 \times 10^{12}$ | 82.8 |
| | | RM-based Ranking | $44.5 \times 10^{12}$ | $0.4 \times 10^{12}$ | 85.9 |
| | | Guided-Decoding + RM | $276.2 \times 10^{12}$ | $11.4 \times 10^{12}$ | 86.5 |
| | | Tree-of-Thought + RM | $84.5 \times 10^{12}$ | $3.7 \times 10^{12}$ | 82.0 |
| | | A* Search + RM | $71.6 \times 10^{12}$ | $1.6 \times 10^{12}$ | 83.6 |
| | | Monte Carlo Tree Search + RM | $360.0 \times 10^{12}$ | $16.2 \times 10^{12}$ | 85.5 |
| | | Predictive-Decoding + RM | $182.3 \times 10^{12}$ | $2.9 \times 10^{12}$ | 87.9 |
| | | Predictive-Decoding + RM | $360.5 \times 10^{12}$ | $5.6 \times 10^{12}$ | **89.9** |

**More analysis** including world model, reflection and prompt sensitivity are in Appendix C.

## 5.3 PLANNING AS REWARD OPTIMIZATION

Recall that Predictive-Decoding samples from a distribution to solve the trajectory optimization problem using LLM self-evaluation. Similarly we can define such distribution for any target. The same sampling technique could maximize the objective $\mathcal{J}$ left to right. We examine how Predictive-Decoding performs with two different types of objective:

**Guiding Agent Planning with Heuristics** We employ human-designed reward heuristics $J$ to guide LLM planning, following the approach recommended by Ma et al. (2024). The heuristic rewards are sparse, requiring multiple steps to increase rewards. As shown in Table 7, even without foresight, the reward enhances LLM agent performance. Predictive-Decoding further overcomes the sparsity with foresight and improves the utility of reward functions.

**Guiding LLM Reasoning with Reward Model** We use Math-Shepherd (Wang et al., 2023b) as the reward model for GSM8K. We compare Predictive-Decoding against other reward model based planning methods. Results in Table 8 demonstrate that all algorithms improve by over 10% with Math-Shepherd. Although strong baselines like Guided-Decoding and MCTS are effective, they require extensive computation. Predictive-Decoding achieves a 1.5% improvement over Guided-Decoding with only 65% of the computation and a 2.4% improvement over MCTS with just 50% of the computation. We further compare sample efficiency in §5.4.

### 5.4 DISCUSSION: INFERENCE SCALING LAW

Snell et al. (2024) recently stressed the importance of inference scaling law, which states that LLM inference could benefit from using more computation to support more expressive algorithms. Wu et al. (2024b) further discuss the efficiency-performance trade-off of different search algorithms. However, it is yet to be discussed if searching is the most efficient and effective choice for LLM reasoning.

We follow setting in §5.3 using reward model Math-Shepherd. In Figure 6, we plot the inference scaling law of various methods and observe that: (i) Sampling-based methods have better computation efficiency compared to search-based methods. However autoregressive generation using simple reward ranking tends to saturate when more computation is available. (ii) Predictive-Decoding which directly samples from optimal solution space achieves better scaling law than all sampling and search baselines. Predictive-Decoding performance also consistently improves with more computation.

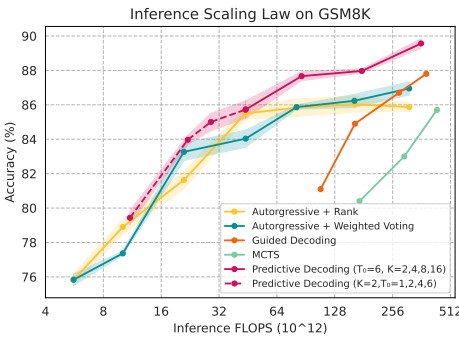

Figure 6: Inference time scaling law on 189 examples sampled from GSM8K. All algorithms use Llama3-8B and Math-Shepherd.

## 6 RELATED WORK

**LLM-based Planning and Reasoning** One major development is the emergence of LLM step-by-step reasoning abilities (Wei et al., 2022). This capability is further enhanced by more expressive inference algorithms like searching (Yao et al., 2022; 2024; Hao et al., 2023; Wang et al., 2023a; Zheng et al., 2023; Xie et al., 2024; Zhao et al., 2024; Sun et al., 2024). However, most of these algorithms are computation-intensive. Recent work discusses the inference time scaling law for LLM reasoning (Wu et al., 2024b; Snell et al., 2024). This highlights the need for an efficient yet global-optimal method. Our work attempts to close this research gap by introducing Predictive-Decoding. Another line of work studies the expressiveness of LLM reasoning. Feng et al. (2024) confirms LLM reasoning's expressiveness equals dynamic programming. Li et al. (2024) discusses Turing completeness in chain-of-thought reasoning. Our theory does not contradict their claims but examines whether any pretrained LLM could find such an optimal solution.

**Model Predictive Control** Model Predictive Control (MPC) is a widely used control strategy that involves solving an optimization problem at each time step by forecasting future results (Witkin & Kass, 1988; Qin & Badgwell, 1997). Similarly, in model-based reinforcement learning (RL), an agent aims to maximize future success by using a dynamics model to simulate state transitions (Silver et al., 2017; Ha & Schmidhuber, 2018; Anthony et al., 2017; Racanière et al., 2017; Nagabandi et al., 2018). These algorithms enable agents to think ahead, envisioning the outcomes of various potential actions, and making deliberate selections among alternatives. Our method follows this line of work.

## 7 CONCLUSIONS AND LIMITATION

In this paper, we analyzed the limitations and potential of Large Language Models (LLMs) in planning and reasoning tasks, particularly focusing on their myopic tendencies. We introduced Predictive-Decoding, which employs Model Predictive Control to enhance non-myopic planning, significantly improving performance across various domains. Our experiments confirmed Predictive-Decoding's effectiveness in boosting planning accuracy and computational efficiency. These results open promising avenues for future research into optimizing LLM reasoning and incorporating foresight into decision-making, paving the way for more robust and efficient LLM applications.

The main limitation of our work is that we are targeted towards improving reasoning and planning at inference times, without exploring how our method could aid in generating training data—an advantage of search-based methods. Additionally, Predictive-Decoding is limited to tasks with clear reasoning steps, like math or coding. While it potentially could aid global awareness in free-form generation, such as long-context tasks, we leave this for future research.

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

# A PSEUDO CODE

---
**Algorithm 1** Predictive-Decoding for Planning

---
**Input:** `prompt`, the language model, maximum number of iterations $T$, sampling number $K$, sampling temperature $\tau$, environment `env`, rollout length $T_0$.
**Output:** Action sequence $a'_0, \ldots a'_T$.

Set $\mathbf{s}_0 \leftarrow$ Initialize `env`, `finish` $\leftarrow$ False
**for** $t = 1, 2, \ldots, T$ **do**
    ▷ Sample Foresight.
    Input `prompt` and $\mathbf{s}'_0, a'_0, \ldots, \mathbf{s}'_t$ to language model `context`;
    **for** $k = 1, 2, \ldots, K$ **do** // In parallel
        Sample $a^k_t, \mathbf{s}^k_{t+1}, a^k_{t+1}, \ldots, a^k_{t+T_0}, \mathbf{s}_{t+T_0+1} \sim P^{\text{LLM}}(\cdot \mid \text{context})$;
        $P_k \leftarrow P^{\text{LLM}}(\mathbf{a}^k_{\geq t}, \mathbf{s}^k_{>t} \mid \text{context})$;
        $w_k \leftarrow \exp(P_k/\tau)$;
    **end for**
    ▷ Re-sample based on foresight.
    Sample $j \sim$ Categorical $\left( \frac{w_1}{\sum_{k=1}^K w_k}, \cdots, \frac{w_K}{\sum_{k=1}^K w_k} \right)$;
    Set $a'_t \leftarrow a^j_t$;
    ▷ Takes the action $a'_t$
    Update $\mathbf{s}_{t+1}$, `finish` $\leftarrow$ Execute $a'_t$ in environment `env`;
    **break if** `finish` is True;
**end for**
**Return** the action sequence $a'_0, \ldots a'_T$.

---

# B THEORETICAL RESULTS

## B.1 SAMPLE EFFICIENCY OF MODEL PREDICTIVE CONTROL

In this section, we explain in detail why model predictive control has the potential to improve sample efficiency in solving trajectory optimization problems.

We begin by introducing a stochastic setting, where trajectory $\tau = (a_0, a_1, \ldots, a_T)$ is a chain of actions that could take values in set $\mathcal{A}$. A policy model models the distribution of trajectories as $\pi_\theta(\tau)$, denoting the probability that the policy model samples the trajectory.

For the best scenario, the policy model is perfect and the sample $\tau' \sim \pi_\theta(\tau)$ maximizes the objective $\mathcal{J}(\tau)$ and $\tau'$ is the best solution. However, in most cases our LLM is not the perfect policy model and prone to make errors during sampling. Instead, we sample multiple solutions $\tau_1, \tau_2, \ldots, \tau_k \sim \pi_\theta(\tau)$ and selects the best solution $\tau' = \max_{i=1,\ldots,k} \mathcal{J}(\tau_i)$. A different method is using model predictive control during sampling. We thereby discuss the number of $k$ necessary to obtain the correct $\tau'$.

**Proposition B.1.** *Given the optimal trajectory* $\tau' = (a'_0, \ldots, a'_T)$ *that maximizes* $\mathcal{J}$. $\tau^1, \tau^2, \ldots, \tau^{k|\mathcal{A}|} \sim \pi_\theta(\tau)$. *The best* $a^r_0$ *is the first step of the trajectory* $\tau^j$ *that maximizes* $\mathcal{J}$.

*If* $a^m_0$ *satisfies:*

$$a^m_0 = \arg\max_{a_0} \left[ \max_{i=1\ldots k} \mathcal{J}(a_0, \tau^i_{\to a_0}) \right], \tau^i_{\to a_0} \sim \pi_\theta(\cdot \mid a_0) \qquad (7)$$

*Then* $P(a^m_0 = a'_0) \geq P(a^j_0 = a'_0)$

*Informal proof*: If $\pi$ has no knowledge of $\mathcal{J}$, then the best case in sampling would be $\tau^1, \tau^2, \ldots, \tau^{k|\mathcal{A}|}$ first sample $a_0$ from $U(a_0)$, then samples $\tau_{\to a_0} \sim \pi_\theta(\cdot \mid a_0)$, in order to gain thorough information about which $a_0$ would be the best for the trajectory. In this case, the sampling process is same as model predictive control and thus the best case results in $P(a^m_0 = a'_0) = P(a^j_0 = a'_0)$. However, when $\pi_\theta$ places more importance on $a_0 \neq a'_0$, i.e. $\exists a_0, \pi(a^*_0) > \pi(a'_0)$, there exists a larger possibility that $a^*_0$ is selected by $J$ instead of $a'_0$ than model predictive control, as let $n$ be the times $a^*_0$ is sampled,

$k > n$, then $E\left[\max_{i=1,\dots,n} \mathcal{J}(a_0^*, \tau_{\to a_0^*}^i)\right] \geq E\left[\max_{i=1,\dots,k} \mathcal{J}(a_0^*, \tau_{\to a_0^*}^i)\right]$. Thus the probability that $a_0^*$ is selected would be larger and $P(a_0^m = a_0') \geq P(a_0^j = a_0')$.

This proposition show that if directly sample solution from an exponential solution space, it would be unlikely to obtain the optimal solution. We also showcase this phenomenon that model predictive control improves sample efficiency than direct sampling through simulated experiments.

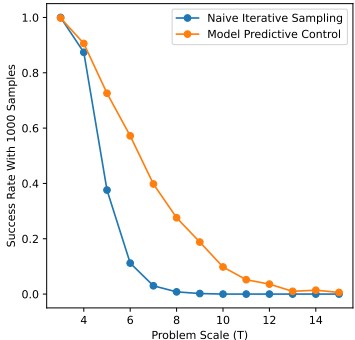

Figure 7: Pass@1000 accuracy v.s. problem scale on the simulation task. $\pi_\theta$ follows a uniform distribution. The performance of iterative sampling decreases significantly when the problem scales up.

Figure 8: Pass@k accuracy on simulation task with different sampling numbers. Larger $\tau$ denotes that the sampling distribution $\pi_\theta$ is close to ground truth model.

We focus on a very simple combinatorial optimization problem, where a list $(a_0, a_1, \dots, a_T)$ satisfies $a_0 \in \mathcal{A}$ takes the value from $|\mathcal{A}|$ integers. Our objective is to find the list of numbers with maximum sum. We control the sample complexity of "random sampling + ranking" and "model predictive control". Results are shown in Figure 7 and Figure 8. On all settings, MPC is more sample efficient than naive iterative sampling. Notably, we alter the prior distribution $\pi_\theta$ such that it samples the number based on value $e^{\text{number}/\tau}$. Larger $\tau$ is closer to uniform distribution yet smaller $\tau$ is closer to the optimal distribution. We can see that a good prior always results in improved performances, and the gap between iterative sampling and MPC is smaller. However, when the $\pi_\theta$ is not good (e.g. using a LLM that is not well-pretrained), the sample efficiency of MPC is much better than iterative sampling.

## B.2 PROOF FOR PROPOSITION 4.1

The distribution that solves the optimization problem in Eq.5 is in the form of:

$$p_\tau(a_t) \propto P^{\text{LLM}}\left(a_t \mid \mathbf{a}_{<t}', \mathbf{s}_{<t}'\right) \exp\left[\mathbb{E}_{\mathbf{a}_{>t}, \mathbf{s}_{>t}} P^{\text{LLM}}\left(a_t, \mathbf{a}_{>t}, \mathbf{s}_{>t} \mid \mathbf{a}_{<t}', \mathbf{s}_{<t}'\right)/\tau\right]. \tag{8}$$

*Proof.* The goal is to solve this optimization problem:

$$a_t' = \arg\max_{a_t} \mathbb{E}_{\mathbf{a}_{>t}, \mathbf{s}_{>t}} P^{\text{LLM}}\left(a_t, \mathbf{a}_{>t}, \mathbf{s}_{>t} \mid \mathbf{a}_{<t}', \mathbf{s}_{<t}'\right).$$

Energy-based Model can transfer the optimization problem into probability distribution, such as,

**Lemma B.2.** *(LeCun et al., 2006) Let $\mathbf{f} = \{f_i\}_{i=1\dots n}$ is a set of evaluation metrics, the distribution $p(x)$ solves the optimization problem $x = \arg\min_{x \in \mathcal{P}} \sum_i \mu_i f_i(x)$, only if $p(x)$ satisfies:*

$$p(x) \propto p_\theta(x) \exp\left[-E_\mu(x)\right], \forall x \in \mathcal{P} \tag{9}$$

*where $\mathcal{P}$ is the support set of probability distribution of $p_\theta$.*

Here $a_t$ is generated by LLM through autoregressive generation, i.e. $a_t \sim P^{\text{LLM}}(a_t \mid \mathbf{a}_{<t}, \mathbf{s}_{<t})$, and the constraint can be rewritten as $f(a_t) = \mathbb{E}_{\mathbf{a}_{>t}, \mathbf{s}_{>t}} P^{\text{LLM}}\left(a_t, \mathbf{a}_{>t}, \mathbf{s}_{>t} \mid \mathbf{a}_{<t}', \mathbf{s}_{<t}'\right)$. Plugging into Eq. 9 and we would yield $pa_t$. $\tau$ can be added as a temperature coefficient and does not lead to difference in optimization. $\square$

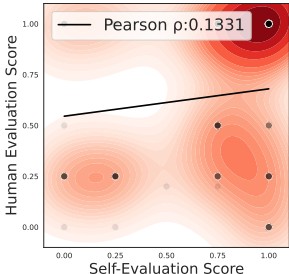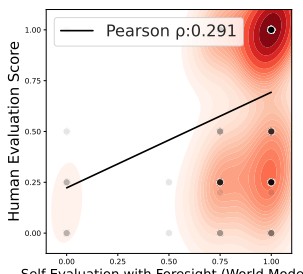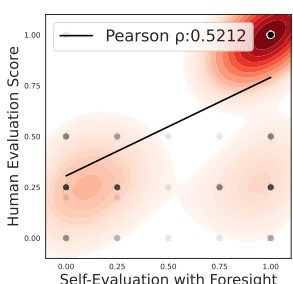

Figure 9: Comparison of GPT-3.5-Turbo Self-Evaluation $\hat{r}(s, a)$ w/o. modeling on 152 trajectory data from AlfWorld. We compare from left-to-right (1) LLM direct self-evaluation based on past trajectory($\rho = 0.133$),(2) LLM first perform model rollout and then perform self-evaluation based on both past and future trajectory($\rho = 0.291$), (3)substituting LLM-as-world-model for an accurate world model for rollout($\rho = 0.521$) against human evaluation. All scores are chosen from {0,0.25,0.5,0.75,1} and we use KDE plots to visualize overlapping discrete points.

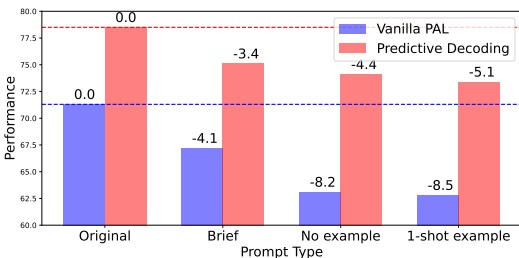

Figure 10: Prompt sensitivity to instruction brevity, as well as the number of in context samples on GSM8K. Both methods use Llama3-8B.

## C  MORE ANALYSIS

***Impact* of Agent World Model Hallucination**   Predictive-Decoding requires the LLM to act as a world model to infer states after each step. However, we often observe hallucinations. For instance, in AlfWorld, when the agent tries to find an apple and place it on desk 1, the imagined state after "open fridge 1" might incorrectly include an apple in the fridge, even if it's not there. This reflects an inherent model bias issue (Kurutach et al., 2018), where the world model fails to accurately represent the environment. Figure 9 shows that this results in lower LLM score accuracy compared to using real environment transitions. Most evaluation errors are false positives, indicating LLM overconfidence in predicting step outcomes. Increasing $\tau$ to 0.05-0.1 helps mitigate this by preventing overfitting.

**Prompt Sensitivity Analysis**   Wang & Zhou (2024) proposes an intereting problem: improving LLM reasoning without prompting. We find Predictive-Decoding can also improves performances and maintain high performance even without in-context samples and detailed instructions. As shown in Figure 10, vanilla generation with LLM show significant drop in performance when the detailed instructionsa are not given, or when there are no in-context examples. Predictive-Decoding shows smaller decline in performance, demonstrating better prompt sensitivity. Moreover, Predictive-Decoding without in-context examples could outperforms vanilla generation with in-context examples. This demonstrate our method also has the potential to improve reasoning without prompting.

**Reflection Analysis**   Shinn et al. (2024); Ma et al. (2024) emphasize the importance of reflection for decision making. However, Predictive-Decoding mainly takes a different approach - it's targeted at avoiding errors, rather than correcting errors. However, for agent tasks that exploration is necessary, the model still needs to adapt plans to new observations. In this section, we anlayze the ability of Predictive-Decoding to perform reflection.

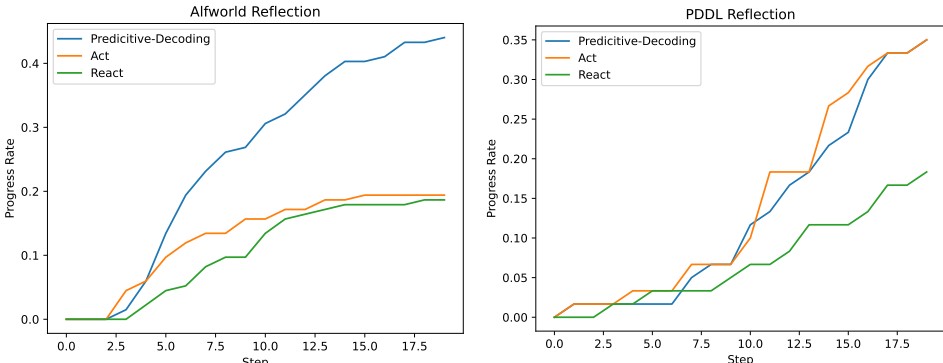

Figure 11: Long-range interaction analysis. Specifically, we report the progress rate w.r.t. step of AlfWorld, PDDL for Llama3.1-70B.

We compare the progress rate of LLM w.r.t to number of interaction steps taken. A model with self-reflection ability would continuously benefit from more interaction steps.

As shown in Figure 11, Predictive-Decoding exhibits continuous improvement across 20 steps. For the AlfWorld task, Predictive-Decoding shows a larger slope across 20 steps compared to Act and ReAct, showcasing improved reflection ability compared with Act and ReAct. For the PDDL task, however, Predictive-Decoding shows comparable self-reflection ability with Act. Notably, on both tasks, Predictive-Decoding shows larger improvement at later steps, which indicates improved long-range reflection ability.

## D    FURTHER DISCUSSION ON RELATED WORK

**Combinatorial Optimization via Sequential Sampling**    Sampling-based methods have been extensively used to solve combinatorial optimization efficiently (Sun & Yang, 2023; Janner et al., 2022; Qin et al., 2022; Du et al., 2024), by constructing an energy-based model where the generative probability reflects the objective. However, these methods often use non-autoregressive sampling, which undermines the Markov property of sequences critical to tasks like planning and language modeling. Autoregressive diffusion models (Wu et al., 2023; Chen et al., 2024a) have been proposed to sample based on optimization constraints while enabling causal generation. However, these methods rely on diffusion training loss and cannot be directly applied to LLMs. In this work, we follow Deng et al. (2020); Ji et al. (2023) and use the sampling-importance-resampling (SIR) technique, which enables us to maintain autoregressive next-token prediction while achieving global optimality. This method can be easily combined with model predictive control and used on any LLM.

## E    CASE STUDY

### E.1    CASE STUDY ILLUSTRATING PREDICTIVE-DECODING

**Case Study for Reasoning**    Here we provide the generations to show how Predictive-Decoding solves a math problem.

> Question: Toulouse has twice as many sheep as Charleston. Charleston has 4 times as many sheep as Seattle. How many sheep do Toulouse, Charleston, and Seattle have together if Seattle has 20 sheep?
>
> def solution():
>
> {' #Toulouse has twice as many sheep as Charleston. Charleston has 4 times as many sheep as Seattle. How many sheep do Toulouse, Charleston, and Seattle have together if Seattle has 20 sheep?': 1.0}

Line 1: #Toulouse has twice as many sheep as Charleston. Charleston has 4 times as many sheep as Seattle. How many sheep do Toulouse, Charleston, and Seattle have together if Seattle has 20 sheep?

{' seattle_sheep = 20': 1.0}

Line 2: seattle_sheep = 20

{' charleston_sheep = seattle_sheep * 4': 0.9767, ' charleston_sheep = 4 * seattle_sheep': 0.0233}

Line 3: charleston_sheep = seattle_sheep * 4

{' toulouse_sheep = charleston_sheep * 2': 0.9377, ' toulouse_sheep = seattle_sheep * 2': 0.0623}

Line 4: toulouse_sheep = charleston_sheep * 2

{' total_sheep = seattle_sheep + charleston_sheep + toulouse_sheep': 1.0}

Line 5: total_sheep = seattle_sheep + charleston_sheep + toulouse_sheep'

{' result = total_sheep': 0.9999}

Line 6: result = total_sheep

{' return result':1.0}

Line 7: return result

Examples in trajectory pool:

Foresight 1 for the first line (composed of 6 steps):

```
{'Action': None, 'Reward': 0.9984305701520592}

{'Action': '    """Toulouse␣has␣twice␣as␣many␣sheep␣as␣Charleston.␣
    ↪ Charleston␣has␣4␣times␣as␣many␣sheep␣as␣Seattle.␣How␣many␣sheep␣do␣
    ↪ Toulouse,␣Charleston,␣and␣Seattle␣have␣together␣if␣Seattle␣has␣20␣
    ↪ sheep?"""\n', 'Reward': 0.9984305701520592}

{'Action': '    seattle_sheep = 20\n',  'Reward': 0.9984305701520592}

{'Action': '    charleston_sheep = seattle_sheep * 4\n', 'Reward': 0
    ↪ .9984305701520592},

{'Action': '    toulouse_sheep = charleston_sheep * 2\n', 'Reward': 0
    ↪ .9984305701520592}

{'Action': '    total_sheep = seattle_sheep + charleston_sheep +
    ↪ toulouse_sheep\n', 'Reward': 0.9984305701520592}
```

Foresight 1 for the third line (composed of 4 steps):

```
{'Action': '    charleston_sheep = seattle_sheep * 4\n', 'Reward': 0
    ↪ .9842750955836792},

{'Action': '    toulouse_sheep = 2 * charleston_sheep\n',  'Reward': 0
    ↪ .9842750955836792}

{'Action': '    total_sheep = seattle_sheep + charleston_sheep +
    ↪ toulouse_sheep\n', 'Reward': 0.9842750955836792}

{'Action': '    result = total_sheep\n', 'Reward': 0.9842750955836792}

{'Action': '    return result\n',  'Reward': 0.9842750955836792}
```

From the case study we could observe that for easier steps, the distribution is concentrated on a single action. This is due to LLM generation is inherently confident on these lines. Our method mainly helps improving the lines where LLM generation is less confident (e.g. Line 3 & 4).

**Case Study for Agent**    In this section we also provide a case study for agent task. Agent tasks are different from reasoning tasks in two ways: (i) Agent tasks make predictions not only based on prior actions, but also on environment observations. Therefore, during foresight, the agent needs to use LLM as a world model (Hao et al., 2023) to infer the next state. (ii) Exploration is crucial for agents. Therefore maintaining stochastic during sampling is important.

Here is the original generation by `GPT-3.5-Turbo`. We can see that this is an example of why focusing solely on historical information can lead to irreversible mistakes and potential planning failures (as mentioned in §1). At the beginning, the agent is exploring the surroundings to look for a bowl. It arrived at shelf 1, where there is only a cellphone and a creditcard. The agent picks up the cellphone despite given instructions and know it needs to find a bowl. As the agent can only carry one thing at a time, it became stuck when it actually finds a bowl and found it has no arm to carry. Therefore, this case results in a planning failure due to the tendency of models to perform history-consistent completion of the next action.

---

Goal: look at bowl under the desklamp.

Observation: You are in the middle of a room. Looking quickly around you, you see a bed 1, a desk 2, a desk 1, a drawer 6, a drawer 5, a drawer 4, a drawer 3, a drawer 2, a drawer 1, a garbagecan 1, a laundryhamper 1, a safe 1, a shelf 6, a shelf 5, a shelf 4, a shelf 3, a shelf 2, and a shelf 1.

Step 1: go to desk 1

Observation: On the desk 1, you see a desklamp 1, a keychain 3, a mug 2, a pen 2, a pen 1, and a pencil 2.

Step 2: take desklamp 1 from desk 1

Observation: Nothing happens.

Step 3: examine desk 1

Observation: On the desk 1, you see a desklamp 1, a keychain 3, a mug 2, a pen 2, a pen 1, and a pencil 2.

Step 4: go to shelf 1

Observation: On the shelf 1, you see a cellphone 3, and a creditcard 2.

Step 5: take cellphone 3 from shelf 1

Observation: You pick up the cellphone 3 from the shelf 1.

Step 6: go to desk 1

Observation: On the desk 1, you see a desklamp 1, a keychain 3, a mug 2, a pen 2, a pen 1, and a pencil 2.

...

---

Below is the revised generation provided by Predictive-Decoding. After incorporating foresight and resampling actions, the task could be finished by the same LLM.

---

Goal: look at bowl under the desklamp.

Observation: You are in the middle of a room. Looking quickly around you, you see a bed 1, a desk 2, a desk 1, a drawer 6, a drawer 5, a drawer 4, a drawer 3, a drawer 2, a drawer 1, a garbagecan 1, a laundryhamper 1, a safe 1, a shelf 6, a shelf 5, a shelf 4, a shelf 3, a shelf 2, and a shelf 1.

{'go to desk 1': 0.9975, 'go to desk 2': 0.0025}

---

Step 1: go to desk 1

Observation: use desklamp 1

{'use desklamp 1': 0.4530, 'examine desklamp 1': 0.5469, 'examine desk 2': 8.551e-06}

Step 2: use desklamp 1

Observation: You turn on the desklamp 1.

{'examine desk 1': 0.9822, 'examine desklamp 1': 0.0177}

Step 3: examine desk 1

Observation: On the desk 1, you see a desklamp 1, a keychain 3, a mug 2, a pen 2, a pen 1, and a pencil 2.

{'examine desklamp 1': 0.5711, 'go to desk 2': 0.4289}

Step 4: examine desklamp 1

Observation: Nothing happens.

{'go to shelf 1': 0.9974, 'go to desk 2': 0.00258}

Step 5: examine shelf 1

Observation: On the shelf 1, you see a cellphone 3, and a creditcard 2.

{'go to desk 2': 0.8146, 'go to shelf 2': 0.1853}

Step 6: go to desk 2

Observation: On the desk 2, you see a alarmclock 1, a bowl 1, and a cd 3.

{'take bowl 1 from desk 2': 0.9626, 'examine bowl 1': 0.03733}

Step 7: take bowl 1 from desk 2

Observation: You pick up the bowl 1 from the desk 2.

{'go to desk 1': 1.0}

Step 8: go to desk 1

Observation: On the desk 1, you see a desklamp 1, a keychain 3, a mug 2, a pen 2, a pen 1, and a pencil 2. The task is finished.

Examples in trajectory pool:

Foresight 1 for the first step (composed of 5 steps):

```
{'Action': None, 'Observation': 'You are in the middle of a room.
    ↪ Looking quickly around you, you see a bed 1, a desk 2, a desk 1, a
    ↪ drawer 6, a drawer 5, a drawer 4, a drawer 3, a drawer 2, a drawer
    ↪ 1, a garbagecan 1, a laundryhamper 1, a safe 1, a shelf 6, a shelf
    ↪ 5, a shelf 4, a shelf 3, a shelf 2, and a shelf 1.', 'Reward': 0
    ↪ .9281983462766966}

{'Action': 'go to desk 1', 'Observation': 'On the desk 1, you see a book
    ↪ 1, a computer 1, a desklamp 1, a notebook 1, and a pen 1.',
    ↪ 'Reward': 0.9281983462766966}

{'Action': 'examine desklamp 1', 'Observation': 'The desklamp 1 is
    ↪ currently turned off.', 'Reward': 0.9281983462766966}

{'Action': 'use desklamp 1', 'Observation': 'You turn on the desklamp 1
    ↪ .', 'Reward': 0.9281983462766966}
```

```
{'Action': 'examine desk 1',  'Observation': 'On the desk 1, you see a
    ↪ book 1, a computer 1, a desklamp 1, a notebook 1, a pen 1, and
    ↪ nothing is under the desklamp 1.', 'Reward': 0.9281983462766966}

{'Action': 'go to desk 2', 'Observation': 'On the desk 2, you see a bowl
    ↪ 1, a cup 1, and a desklamp 2.', 'Reward': 0.9281983462766966}
```

Foresight 1 for the second step (composed of 5 steps):

```
{'Action': 'go to desk 1','Observation': 'On the desk 1, you see a
    ↪ desklamp 1, a keychain 3, a mug 2, a pen 2, a pen 1, and a pencil 2
    ↪ .', 'Reward': 0.9279943292829683}

{'Action': 'use desklamp 1',  'Observation': 'You turn on the desklamp 1
    ↪ .', 'Reward': 0.9279943292829683}

{'Action': 'examine desk 1', 'Observation': 'On the desk 1, you see a
    ↪ desklamp 1, a keychain 3, a mug 2, a pen 2, a pen 1, a pencil 2,
    ↪ and a drawer 1.', 'Reward': 0.9279943292829683}

{'Action': 'open drawer 1', 'Observation': 'You open the drawer 1.
    ↪ Inside, you see a bowl 1.', 'Reward': 0.9279943292829683}

{'Action': 'take bowl 1 from drawer 1', 'Verified': None, 'Observation':
    ↪ 'You pick up the bowl 1 from the drawer 1.', 'Reward': 0
    ↪ .9279943292829683}

{'Action': 'put bowl 1 on desk 1', 'Observation': 'You put the bowl 1 on
    ↪ the desk 1, under the desklamp 1.', 'Reward': 0.9279943292829683}
```

Foresight 2 for the second step (composed of 5 steps, lower quality):

```
{'Action': 'go to desk 1', 'Observation': 'On the desk 1, you see a
    ↪ desklamp 1, a keychain 3, a mug 2, a pen 2, a pen 1, and a pencil 2
    ↪ .', 'Reward': 0.7375680280039042}

{'Action': 'examine desklamp 1',  'Observation': 'The desklamp 1 is
    ↪ currently turned off.', 'Reward': 0.7375680280039042}

{'Action': 'use desklamp 1', 'Observation': 'You turn on the desklamp 1
    ↪ .', 'Reward': 0.7375680280039042}

{'Action': 'go to desk 2', 'Observation': 'On the desk 2, you see a book
    ↪ 1, a bowl 1, a clock 1, and a stapler 1.', 'Reward': 0
    ↪ .7375680280039042}

{'Action': 'examine bowl 1', 'Observation': 'You are now looking at the
    ↪ bowl 1 under the light from the desklamp 1.', 'Reward': 0
    ↪ .7375680280039042}

{'Action': 'inventory',  'Observation': 'You have nothing in your
    ↪ inventory.', 'Reward': 0.7375680280039042}
```

E.2 FAILURE CASE ANALYSIS FOR PREDICTIVE-DECODING

Here we analyze typical failure cases using Predictive-Decoding. We mainly analyze examples where Predictive-Decoding could be sub-optimal (perform worse than autoregressive generation).

**Repetition of in-context examples** Several failed cases in the MATH dataset shows repetition of in-context examples in the answer of an unrelated question. This is possibly due to that improving

overall confidence of generation would sometimes result in repetition, as the model yields a high confidence when copying from in-context examples. This constitutes 1% of the wrong examples on MATH. Below is an example of this kind of error:

Problem: 3 cards are chosen at random from a standard 52-card deck. What is the probability that they form a pair? (A 3-card hand is a 'pair' if two of the cards match in rank but the third card is different. For example, 668 is a pair, but 999 is not.)

```
import itertools

import numpy as np

from sympy import symbols, simplify

def solution():

    digits = [1, 3, 4, 5, 9]

    list_of_integers = []

x = symbols('x')

expr = 3*(x**2 - x**3 + x) + 3*(x + 2*x**3 - 3*x**2 + 3*x**5 + x**3) - 5
    ↪ *(1 + x - 4*x**3 - x**2)

simplified_expr = simplify(expr)

x3_coefficient = simplified_expr.as_coefficients_dict()[x**3]

result = x3_coefficient

return result
```

**Token-level Spelling mistakes**    A few failed cases in GSM8K dataset shows minor spelling mistakes in variables, resulting in failed execution. For example, **raymond jewels** v.s. **raymond jews**. This is due to the averaged probability calculation at the step level may ignore token level mistakes. However, as LLM rarely makes these mistakes, these cases are rare. Below is an example of this kind of error:

```
def solution():
    """Siobhan␣has␣2␣fewer␣jewels␣than␣Aaron.␣Aaron␣has␣5␣more␣jewels␣
        ↪ than␣half␣of␣Raymond's␣jewels.␣If␣Raymond␣has␣40␣jewels,␣how␣
        ↪ many␣jewels␣does␣Siobhan␣have?"""

    raymond_jewels = 40

    aaron_jewels = (raymond_jews) / 2 + 5

    siobhan_jewels = aaron_jewels - 2

    result = siobhan_jewels

    return result
```

## F    EXPERIMENT PROMPTS AND PARAMETERS

### F.1    IMPLEMENTATIONS AND HYPERPARAMETERS

**Large Language Models**    Table 9 show the source of LLMs we used in our experiment. We use 1-4 A100s to launch the LLMs with vLLM. The vLLM configuration uses standard configs and use `fp16` for all models.

**Baseline Implementations**

| Model Name | Model Code/API |
|---|---|
| GPT-3.5-Turbo (Achiam et al., 2023) | Azure api: `gpt-35-turbo` |
| Mistral-7b  (Jiang et al., 2023) | `mistralai/Mistral-7B-v0.3` |
| Llama3-8b (Dubey et al., 2024) | `meta-llama/Meta-Llama-3-8B-Instruct` |
| Llama3.1-70b (Dubey et al., 2024) | `meta-llama/Llama-3.1-70B-Instruct` |
| Deepseek-Coder-6.7b (Guo et al., 2024) | `deepseek-ai/deepseek-coder-6.7b-instruct` |

Table 9: Model code/API of our evaluated models.

- **PAL**: We use the setting of first generating the entire code and then obtain the answers through a Python environment. We use the implementation from Guo et al. (2024).

- **Beam Search**: We use the standard implementation provided by vLLM.

- **Guided Decoding**: We use a widely used implementation from LLM-Reasoners (Hao et al., 2023).

- **Self-Consistency**: We reimplement Self-Consistency following (Wang et al., 2022). For PAL generation, we only weight the answers from successful executed code.

- **Self Infilling**: We use Self Infilling implementation following  Zheng et al. (2023). However this method requires suffix tokens, which is not available for Llama3. Instead, we first generate an entire code and then use logp to heuristically find the start of suffix.

- **Act**: We use the implementation from AgentBoard (Ma et al., 2024).

- **ReAct**: We reimplement ReAct following Yao et al. (2022). Note that we use the setting that LLM could freely choose to alternate between thinking and acting. We find this performs more stably compared to compulsory thinking at each step.

- **MCTS**: We use the MCTS implementation from LLM-Reasoners (Hao et al., 2023). However, the original implementation for GSM8K uses subquestions, which is different from other baselines. We unify each step as COT format.

**Hyperparameters for Main Experiments**   Table 10 shows the hyperparameters used for main experiments for Predictive-Decoding. Most tasks prefer a moderate LLM generation temperature and very low selection temperature. However, we find that Deepseek-Coder prefer otherwise. This is due to Deepseek-Coder generation is more random, making accurate sampling of foresight difficult, therefore we need smaller $\tau$ to prevent overfitting. The FLOPS for Predictive-Decoding is roughly $O(KT_0)$, however we use parallel accelerated implementation and the actual speed is roughtly $O(T_0)$.

For math and coding tasks $N = 1$ Sampling baselines (PAL, COT, Self Infilling) use $\alpha = 0.6$, $N > 1$ use $\alpha = 1.0$ if otherwise stated. For agent tasks, Act and ReAct use temperature $\alpha = 0.0$.

Beam Search uses the same beam size and best-of-n as $N$, $N = 8$ for math tasks and $N = 10$ for coding tasks. The FLOPS for beamsearch is roughly $O(N^2)$ under this implementation.

Guided Decoding uses the standard hyperparameters from LLM-Reasoners implementation: beam size = 5, depth limit = 16, number of generations for each step = 8, beam search temperature 0.5, and reject minimum reward = 0.6.

## F.2   PROMPTS

---

**Prompt Details for Coding Tasks**

**System Prompt**
Finish writing the python function. You will only write code blocks. Write # finish after the last line of the function.

- - - - - - - - - - - - - - - - - - - - - - - - - - - - - - - - - - - - - - - - - - - - - - - - - - - - - - - -
**Instruction**
{Instruction of the problem}

---

Table 10: Hyperparameters for Predictive-Decoding main experiments.

| Method | Model | Task | Hyperparameters |
|---|---|---|---|
| Predictive-Decoding | Llama3-8B | MATH | $\alpha = 1.0, \tau = 0.01, K = 8, T_0 = 6$ |
| | | GSM8K | $\alpha = 1.0, \tau = 0.01, K = 8, T_0 = 6$ |
| | | HumanEval | $\alpha = 0.3, \tau = 0.05, K = 8, T_0 = 6$ |
| | | MBPP | $\alpha = 1.0, \tau = 0.1, K = 8, T_0 = 6$ |
| | Mistral-v0.3 | MATH | $\alpha = 1.0, \tau = 0.01, K = 8, T_0 = 6$ |
| | | GSM8K | $\alpha = 1.0, \tau = 0.01, K = 8, T_0 = 6$ |
| | Deepseek-Coder | HumanEval | $\alpha = 0.4, \tau = 1.0, K = 8, T_0 = 6$ |
| | | MBPP | $\alpha = 0.4, \tau = 1.0, K = 8, T_0 = 6$ |
| | Llama3.1-70B | Alfworld | $\alpha = 1.0, \tau = 0.01, K = 8, T_0 = 5$ |
| | | PDDL | $\alpha = 1.0, \tau = 0.01, K = 8, T_0 = 5$ |
| | GPT-35-Turbo (azure API) | Alfworld | $\alpha = 0.6, \tau = 0.01, K = 8, T_0 = 5$ |
| | | PDDL | $\alpha = 0.8, \tau = 0.05, K = 8, T_0 = 5$ |

---

**Prompt Details for GSM8K-COT style**

**System Prompt**
You will solve math problems following examples.

- - - - - - - - - - - - - - - - - - - - - - - - - - - - - - - - - - - - - - - - - - - - - - - - -

**Examples**
Q: There are 15 trees in the grove. Grove workers will plant trees in the grove today. After they are done, there will be 21 trees. How many trees did the grove workers plant today?
A: There are 15 trees originally. Then there were 21 trees after some more were planted. So there must have been 21 - 15 = 6. The answer is 6.
Q: If there are 3 cars in the parking lot and 2 more cars arrive, how many cars are in the parking lot?
A: There are originally 3 cars. 2 more cars arrive. 3 + 2 = 5. The answer is 5.
...
Q:...
A:...

- - - - - - - - - - - - - - - - - - - - - - - - - - - - - - - - - - - - - - - - - - - - - - - - -

**Instruction**
Solve this problem following previous examples: Q: {question}

---

**Prompt Details for GSM8K-PAL style**

**System Prompt**
You will write python program to solve math problems. You will only write code blocks.

- - - - - - - - - - - - - - - - - - - - - - - - - - - - - - - - - - - - - - - - - - - - - - - - -

**Examples**
Q: Olivia has $23. She bought five bagels for $3 each. How much money does she have left?
# solution in Python:

```python
def solution():
    """Olivia has $23. She bought five bagels for $3 each.
    How much money does she have left?"""
    money_initial = 23
    bagels = 5
    bagel_cost = 3
    money_spent = bagels * bagel_cost
```

```
        money_left = money_initial - money_spent
        result = money_left
        return result
```

Q: ...
# solution in Python:
... - - - - - - - - - - - - - - - - - - - - - - - - - - - - - - - - - - - - - - - - - - - - - - - -
**Instruction**
Solve this problem following previous examples: Q: {question}
# solution in Python:

```
def solution():
```

---

**Prompt Details for MATH-PAL style**

**System Prompt**
You will write python program to solve math problems. You will only write code blocks.

- - - - - - - - - - - - - - - - - - - - - - - - - - - - - - - - - - - - - - - - - - - - - - - - - -

**Examples**
Let's write python function to solve math problems. You must return the executed result at the end of the function in float. If the final result is an expression, return it in LaTeX in simplest form. You can only write a single function.
Here are some examples:
Question: Find the coefficient of $x^3$ when $3(x^2 - x^3 + x) + 3(x + 2x^3 - 3x^2 + 3x^5 + x^3) - 5(1 + x - 4x^3 - x^2)$ is simplifie.
solution in Python:

```
from sympy import symbols, simplify

def solution():
    x = symbols('x')
    expr = 3*(x**2 - x**3 + x) + 3*(x + 2*x**3 - 3*x**2
    \+ 3*x**5 + x**3) - 5*(1 + x - 4*x**3 - x**2)
    simplified_expr = simplify(expr)

    x3_coef = simplified_expr.as_coefficients_dict()[x**3]
    result = x3_coef
    return result
```

Q: ...
solution in Python:
... - - - - - - - - - - - - - - - - - - - - - - - - - - - - - - - - - - - - - - - - - - - - - - - -
**Instruction**
Solve this problem following previous examples: Q: {question}
solution in Python:

```
def solution():
```

---

**Prompt Details for AlfWorld**

**System Prompt**
You are a helpful assistant. Generate your next step of action after Action. Action must not be empty. e.g. Action: put down cup.
- - - - - - - - - - - - - - - - - - - - - - - - - - - - - - - - - - - - - - - - - - - - - - - - - -

**Instruction**
Your task is to interact with a virtual household simulator to accomplish a specific task. With each interaction, you will receive an observation. Your role is to decide on an action based on the observation. Please ensure that any objects ('obj') and receptacles ('recep') you mention in your response are present in the observation provided.
Here are the available actions you can take:
- take obj from recep
- put obj in/on recep
- open recep
- close recep
- toggle obj/recep
- clean obj using recep
- cool obj using recep
- heat obj using recep
- inventory
- examine recep/obj
- go to recep

- - - - - - - - - - - - - - - - - - - - - - - - - - - - - - - - - - - - - - - - - - - - - - - - - - - - - - - - -
**Examples**
Your task is to: look at statue under the desklamp.
You are in the middle of a room. Looking quickly around you, you see a coffeetable 1, a diningtable 1, a drawer 4, a drawer 3, a drawer 2, a drawer 1, a dresser 1, a garbagecan 1, a sidetable 2, a sidetable 1, and a sofa 1.
Action: go to dresser 1
Observation: On the dresser 1, you see a cellphone 3, a newspaper 2, a statue 1, and a television 1.
Action: take statue 1 from dresser 1
Observation: You pick up the statue 1 from the dresser 1.
Action: go to sidetable 1
Observation: On the sidetable 1, you see nothing.
Action: go to sidetable 2
Observation: On the sidetable 2, you see a desklamp 3, a newspaper 1, and a statue 2.
Action: use desklamp 3
Observation: You turn on the desklamp 3.

---

Prompt Details for PDDL

**System Prompt**
You are a master in planning. Generate your next step of action after Action. Action must not be empty. e.g. Action: put down cup.

- - - - - - - - - - - - - - - - - - - - - - - - - - - - - - - - - - - - - - - - - - - - - - - - - - - - - - - - -
**Instruction**
The robot has four actions: pickup, putdown, stack, and unstack. The domain assumes a world where there are a set of blocks that can be stacked on top of each other, an arm that can hold one block at a time, and a table where blocks can be placed.
The actions defined in this domain include:
pickup `<block>`: allows the arm to pick up a block from the table if it is clear and the arm is empty. After the pickup action, the arm will be holding the block, and the block will no longer be on the table or clear.
putdown `<block>`: allows the arm to put down a block on the table if it is holding a block. After the putdown action, the arm will be empty, and the block will be on the table and clear.
stack `<block>` `<block>`: allows the arm to stack a block on top of another block if the arm is holding the top block and the bottom block is clear. After the stack action, the arm will be empty, the top block will be on top of the bottom block, and the bottom block will no longer be clear.

unstack `<block>` `<block>`: allows the arm to unstack a block from on top of another block if the arm is empty and the top block is clear. After the unstack action, the arm will be holding the top block, the top block will no longer be on top of the bottom block, and the bottom block will be clear.

- - - - - - - - - - - - - - - - - - - - - - - - - - - - - - - - - - - - - - - - - - - - - - -

**Examples**
Goal: The goal is to satisfy the following conditions: b1 is on b2., b2 is on b3. Observation: b1 is on the table. b2 is on the table. B3 is on the table. Robot arm is empty. The b1 is clear. The b2 is clear. The b3 is clear.
Action: pickup b2
Observation: b1 is on the table. B2 is on the table. The b1 is clear. The b3 is clear. You are holding b2.
Action: stack b2 b3
Observation: b1 is on the table. b1 is on b2. B3 is on the table. Robot arm is empty. The b1 is clear. The b2 is clear.
Action: pickup b2.
Observation: The action is not valid and therefore takes no effect. Please remember to satisfy the restriction of actions. You can also check valid actions.
Action: check valid actions.
Observation: valid actions are: pickup b2, unstack b1 b2.
Action: pickup b1
Observation: b2 is on b3. B3 is on the table. Robot arm is empty. The b2 is clear. You are holding b1.
Action: stack b1 b2
Observation: b1 is on b2. b2 is on b3. B3 is on the table. Robot arm is empty. The b1 is clear. The goal is satisfied.

