# OpenReview forum: "Non-myopic Generation of Language Models for Reasoning and Planning"
_ICLR.cc/2025/Conference — ICLR 2025 Poster_

### Official Review · Reviewer_4xRU · 2024-10-31

**Soundness:** 2
**Presentation:** 3
**Contribution:** 2
**Rating:** 6
**Confidence:** 4

**Summary:**

This paper presents "Predictive-Decoding," a novel optimal-control approach that enhances Large Language Models' planning and reasoning capabilities by leveraging Model Predictive Control to overcome the myopic limitations of autoregressive decoding. The method reweights LLM distributions based on foresight trajectories, demonstrating improved performance in math, coding, and agent-based tasks while being computationally more efficient than existing search-based methods.

**Strengths:**

- The framework can work on different kinds of tasks.
- The method makes sense.
- Attempt to fair comparison on the effort to the number of tokens.

**Weaknesses:**

- The proposed method is not particularly fancy; the Lookahead Search in [1] presented an almost identical approach.
- The paper lacks comparison with the latest search algorithms, such as BFS-based TOT and A*-based Q*[2]. The authors should have included comparisons with these SOTA algorithms in the experimental section.
- The paper needs more discussion on results based on different rewards (Self reflection/Reward model). There seems to be a lack of details regarding the evaluation steps in the search process.
- Many existing papers have suggested that LLMs do not inherently possess self-reflection capabilities. This paper lacks detailed experimental discussion regarding self-feedback abilities.

[1]Scaling LLM Test-Time Compute Optimally can be More Effective than Scaling Model Parameters

[2] Q*: Improving Multi-step Reasoning for LLMs with Deliberative Planning

**Questions:**

- Differentiate between the mentioned Lookahead Search and explain what makes Predictive-Decoding unique.
- Additional experimental results are needed: provide more detailed comparison with TOT and Q*.
- More specifics about the evaluation process during decoding.

---

> ### Author Response · Authors · 2024-11-21
> **Response to Reviewer 4xRU**
>
> Thank you for your careful review and constructive comments.
>
> At the high level, we would like to first clarify several terms. Predictive-Decoding is a search-free algorithm, in that for each step we directly generate the next action and do not keep other options for backtracking as in the search-based algorithms.
> Furthermore, self-reflection represents a separate direction from our experiment settings, as we aim to directly improve planning accuracy by avoiding mistakes, rather than incorporating reflections to fix mistakes. These two methods in improving planning are distinct from each other
>
> These guide our methodology and experimental design. We focus on evaluating direct action generation quality rather than advancing search methods or reflection-based improvements.
>
> We have revised our manuscript to incorporate your valuable suggestions, including:
> - add baselines TOT and A* search in Table 8.
> - add more details of the evaluation process in Section 4.
> - add a self-reflection ability analysis in Appendix C.
>
> -----
> We address the concerns below:
>
> > **W1&Q1: The proposed method is not particularly fancy; the Lookahead Search in [1] presented an almost identical approach. Differentiate between the mentioned Lookahead Search and explain what makes Predictive-Decoding unique.**
>
> Lookahead Search, introduced in [1], primarily performs rollouts 𝑁 steps ahead and uses a process reward model to score the lookahead generation. This score is then used to guide beam search. Predictive-Decoding incorporates several design elements different from Lookahead Search:
> - We use a sampling-resampling algorithm to select an action, which allows for temperature configurations to implement trade-offs between more local or global generations.
> - Predictive-Decoding is search-free, selecting only one action at each step. This aligns with the principles of model predictive control in its design, whereas Lookahead Search, as mentioned in the paper, is representative of MCTS-style methods.
> - Additionally, Predictive-Decoding introduces trajectory recycling to improve sampling efficiency.
>
>
> Moreover, we observed an interesting difference in the experimental findings between the two papers. As shown in Figure 3 of [1], Lookahead Search exhibits worse inference scaling relative to beam search and weighted best-of-N. In contrast, our experiments in Figure 6, which also analyze inference scaling laws using the same reward model, Math-Shepherd, reveal different trends. Our method demonstrates strong performance-efficiency trade-offs under similar experimental conditions. While a direct implementation comparison is not feasible at this time (we are awaiting an open-source implementation of Lookahead Search), these differing results suggest potential advantages in our design approach.
>
> Although incorporating foresight is also explored in contemporary work, our method is unique in proposing a well-performing recipe for non-myopic generation.
>
> [1] Scaling LLM Test-Time Compute Optimally can be More Effective than Scaling Model Parameters

---

> > ### Author Response · Authors · 2024-11-21
> > **Response to Reviewer 4xRU**
> >
> > >  **W2&Q2: The paper lacks comparison with the latest search algorithms, such as BFS-based TOT and A star-based Q star[2].**
> >
> > We add additional experiments for TOT and A* search, and add them to Table 8 in the revised manuscript.
> > TOT uses a DFS-based implementation based on the LLM-Reasoners Implementation. A* follows [1] in calculating the utility function, as [2] requires training a Q* function. Notably, we implement A* by ourself, as previous work using A* search are all not open source (including [1] [2] [3]). Results are shown below.
> >
> > | Method                       | Inference FLOPS | Reward FLOPS | GSM8K |
> > |------------------------------|-----------------|--------------|-------|
> > | Autoregressive               | $5.6 × 10^{12}$     | $0.0 × 10^{12}$  | 70.4  |
> > | RM-weighted Self-Consistency | $44.5 × 10^{12}$    | $0.4 × 10^{12}$  | 82.8  |
> > | RM-based Ranking             | $44.5 × 10^{12}$    | $0.4 × 10^{12}$  | 85.9  |
> > | Guided-Decoding + RM         | $276.2 × 10^{12}$   | $11.4 × 10^{12}$ | 86.5  |
> > | Monte Carlo Tree Search + RM | $360.0 × 10^{12}$   | $16.2 × 10^{12}$ | 85.5  |
> > | Tree of Thought + RM         | $84.5 × 10^{12}$    | $3.7 × 10^{12}$  | 82.0  |
> > | A* Search + RM               | $71.6 × 10^{12}$    | $1.6 × 10^{12}$  | 83.6  |
> > | Predictive-Decoding + RM     | $182.3 × 10^{12}$   | $2.9 × 10^{12}$  | 87.9  |
> > | Predictive-Decoding + RM     | $360.5 × 10^{12}$   | $5.6 × 10^{12}$  | **89.9**  |
> >
> > Predictive-Decoding still achieves superior performance. We will add additional inference compute scaling results for these two baselines in future versions. Aside from Tree-of-Thought and A*, we have also compared with Guided-Decoding (Step-wise Beam search) and MCTS, which are also SOTA search baselines.
> >
> > [1] "Toolchain*: Efficient action space navigation in large language models with a* search." ICLR (2023).
> >
> > [2] "Q*: Improving multi-step reasoning for llms with deliberative planning."
> >
> > [3] "Litesearch: Efficacious tree search for llm."

---

> > > ### Author Response · Authors · 2024-11-26
> > >
> > > Dear Reviewer 4xRU,
> > >
> > > We would appreciate it if you could let us know whether our response has adequately addressed your concerns and questions. We remain available to address any further questions you may have.
> > >
> > > Thank you once again for your time and effort!

---

> ### Author Response · Authors · 2024-11-21
> **Response to Reviewer 4xRU**
>
> >**W3&Q3: More specifics about the evaluation process during decoding.**
>
> We tested on two different evaluation settings in this work:
>
> - Using LLM self-estimation of foresight steps to guide action resampling. This setting uses LLM probability for evaluation, which calculates the length normalized probability of all tokens within current step and T0 future steps. (Table 3-5)
>
> - Using an external reward to guide generation. We tested two different kinds of reward, designed-heuristics for agent tasks(Table 7), as well as a reward model, math-shepherd for math tasks (Table 8 & Figure 6).
>
> We revised the manuscript in Section 4 to clarify the evaluation process.
>
> >**W4: Many existing papers have suggested that LLMs do not inherently possess self-reflection capabilities. This paper lacks detailed experimental discussion regarding self-feedback abilities.**
>
> As aforementioned, our work is not focused on improving self-reflection capabilities of LLMs. However, for agent tasks, the model indeed involves taking observation and error messages from the environment and fixing the plans. We add an analysis of step-level self-reflection for AlfWorld and PDDL in the revised manuscript Appendix C, following [1][2].
>
> As shown in Figure 11, Predictive-Decoding exhibits continuous improvement across 20 steps. For the AlfWorld task, Predictive-Decoding shows a larger slope across 20 steps, showcasing improved self-reflection ability compared with Act and ReAct. For the PDDL task, however, Predictive-Decoding shows only comparable self-reflection ability with Act. Notably, on both tasks, Predictive-Decoding shows larger improvement at later steps, which indicates improved long-range self-reflection ability.
>
> [1] Xu, Yiheng, et al. "Lemur: Harmonizing natural language and code for language agents." ICLR (2023).
>
> [2] Ma, Chang, et al. "AgentBoard: An Analytical Evaluation Board of Multi-turn LLM Agents." NeurIPS (2024).

---

> ### Comment · Reviewer_4xRU · 2024-11-27
>
> Thank you for the author's response, which addressed most of my concerns. I will increase my score.

---

> > ### Author Response · Authors · 2024-11-27
> > **Thank you for your reply**
> >
> > We sincerely appreciate your valuable feedback!

---

### Official Review · Reviewer_2bKh · 2024-11-01

**Soundness:** 3
**Presentation:** 4
**Contribution:** 4
**Rating:** 8
**Confidence:** 2

**Summary:**

The paper addresses the problem of myopia (short-sightedness) in Large Language Models (LLMs) during reasoning and planning tasks. It proposes a novel method called Predictive-Decoding, which leverages Model Predictive Control (MPC) to mitigate early errors and enhance non-myopic planning. By re-weighting the LLM distributions based on foresight trajectories, the method aims to improve planning accuracy. Extensive experiments demonstrate significant improvements in performance across various tasks, including math problem-solving, coding, and agent-based interactions. The proposed method shows computational efficiency, outperforming search-based baselines with reduced computational resources.

**Strengths:**

- The application of Model Predictive Control to mitigate myopia in LLMs is a novel approach that enhances planning accuracy.
- The paper provides a solid foundation and supports the claims with experimental results.
- The paper is well-organized, with clear explanations of the problem, methodology, and results.

**Weaknesses:**

See questions

**Questions:**

Failure Cases: Can the authors provide more insights into the specific scenarios where Predictive-Decoding may fail or perform sub-optimally? What are the common characteristics of these failure cases?

Broader Applicability: Have the authors considered testing the method on more open-ended tasks, such as advanced data analysis or complex decision-making scenarios?

Parameter Sensitivity: Can the authors provide more details on the sensitivity of the method to different hyperparameters (e.g., foresight length, sampling number)?

---

> ### Author Response · Authors · 2024-11-21
> **Response to Reviewer 2bKh**
>
> Thank you for your constructive review as well as appreciation of our work! Thanks to your proposal, we added a failure case analysis section in Appendix E.2.
>
> We address the questions below:
> > **Q1: Failure Cases: Can the authors provide more insights into the specific scenarios where Predictive-Decoding may fail or perform sub-optimally? What are the common characteristics of these failure cases?**
>
> This is a great suggestion! We add a study of failure cases in the revised manuscript. Please refer to section Appendix E.2
>
> We notice several characteristics of these suboptimal cases where the non-myopic generation is worse than myopic generation:
>
> - *Repetition of in-context examples*: Several failed cases in the MATH dataset exhibit repetition of in-context examples within the answers to unrelated questions. This is possibly because improving the overall confidence of generation can sometimes lead to repetition, as the model tends to yield high confidence when copying from in-context examples. This issue accounts for 1% of the incorrect examples in the MATH dataset. However, this phenomenon does not occur in other datasets, likely due to the inherent difficulty of the MATH task.
>
> - *Token Level Typos*: A few failed cases in GSM8K dataset shows minor spelling mistakes in variables, resulting in failed execution. For example, raymond_jewels -> raymond_jews. This is due to the averaged probability calculation at the step level may ignore token level mistakes. However, as LLM rarely makes these mistakes, these cases are rare.
>
> >**Q2 :Broader Applicability: Have the authors considered testing the method on more open-ended tasks, such as advanced data analysis or complex decision-making scenarios?**
>
> This is great advice. The current decision-making tasks we tested are mostly fully-observable tasks, where the outcomes of actions are easier to anticipate. AlfWorld, however, is a partially-observable task, making the results more difficult to predict, and the world model is more prone to hallucinate. Nevertheless, we find that even with imperfect future estimation (see hallucination analysis in Appendix C), the model still benefits from foresight.
>
> We plan to test more partially-observable agent tasks to analyze the stability of foresight in more complex decision-making scenarios, including tool use and web browsing. We will include these in later revisions, although this may take longer than the rebuttal period.
>
> >**Q3: Parameter Sensitivity: Can the authors provide more details on the sensitivity of the method to different hyperparameters (e.g., foresight length, sampling number)?**
>
> Our method involves altogether four hyperparameters: foresight length, sampling number, as well as two sampling temperature ($\alpha$ and $\tau$).
>
> We provide a detailed analysis of foresight length and sampling number in Table 6 Section 5.2. Generally, this algorithm is sensitive to foresight lengths. Performances on both GSM8K and HumanEval significantly improve with longer foresight, which determines the level of non-myopia in generation.
>
> Increasing sampling number K leads to better performances as it could give a more accurate account of the original generation distribution. However, trajectory recycling significantly improves the sampling efficiency. Thus this hyperparameter affects performance less than foresight length.
>
> For sampling temperature, please refer to our response to **Reviewer fXex Q2**. Generally, action sampling temperature $\tau$ significantly impacts performance, as it also controls the tradeoff between more local/global generation.

---

> > ### Comment · Reviewer_2bKh · 2024-11-24
> >
> > Thank you for the clarifications. I raised my score.

---

> > > ### Author Response · Authors · 2024-11-25
> > > **Thank you for your response**
> > >
> > > We sincerely appreciate your valuable feedback! We remain available to address any further questions you may have.

---

### Official Review · Reviewer_iHG8 · 2024-11-03

**Soundness:** 3
**Presentation:** 3
**Contribution:** 2
**Rating:** 6
**Confidence:** 3

**Summary:**

In this work, authors propose Predictive-Decoding, a training-free approach to improve LLM planning with non-myopic generation, based on their claim that focusing solely on historical information can lead to irreversible mistakes and potential planning failures. Targeting at the myopia issue in language models pretrained with next-token prediction, Predictive-Decoding leverages Model Predictive Control to enhance planning accuracy.

**Strengths:**

1. investigate an important problem, i.e. short-sightedness.
2. Good performance-budget balance in experiments.

**Weaknesses:**

1. case study? (focusing solely on historical information can lead to irreversible mistakes and potential planning failures.)
2. The formulation in lines 126-127 is not a POMDP, and lacks of mapping from global states to local observations.
3. I'm concerned that the author's hypotheses in lines 188-191 are untenable. It seems like saying that "the more confident the answer is for a given model, the more likely it is to be correct." and in this case the greedy answer should usually be selected?

**Questions:**

See the Weaknesses

---

> ### Author Response · Authors · 2024-11-21
> **Response to Reviewer iHG8**
>
> Thank you for your thoughtful review!
>
> We have revised our manuscript to incorporate your valuable suggestions, including:
> - adding a case study to illustrate the specific kind of planning failures in Appendix E.1
> - revising the confusing term POMDP
> -------
>
> > **W1: case study? (focusing solely on historical information can lead to irreversible mistakes and potential planning failures.)**
>
> We add a dedicated example in Appendix E.1 to illustrate the phenomenon. In the case, the agent needs to finish the goal “look at bowl under the desklamp”. At the beginning, the agent is exploring the surroundings to look for a bowl. It arrives at shelf 1, where there is only a cellphone and a creditcard. The agent picks up the cellphone despite given instructions that it needs to find a bowl. As the agent can only carry one thing at a time, it became stuck when it actually finds a bowl and found it has no arm to carry. Therefore, this case results in a planning failure due to the tendency of models to perform history-consistent completion of the next action.
>
> We also show that after using Predictive-Decoding, this task could be successfully completed.
>
> > **W2: The formulation in lines 126-127 is not a POMDP, and lacks of mapping from global states to local observations.**
>
> Thanks for pointing it out. We use the notation "POMDP" following [1][2], which differs from common MDP processes in that the agent takes observations of the world to make decisions, rather than directly relying on the state. This aligns with our setting involving LLM agents; however, we agree that it may seem confusing, especially as we also introduced reasoning tasks.
>
> To address this, we have revised the manuscript to use a more general description, replacing "POMDP" with "decision process."
>
> [1] Xie, Tianbao, et al. "OpenAgents: An Open Platform for Language Agents in the Wild." COLM 2024.
>
> [2] Xiong, Weimin, et al. "Watch Every Step! LLM Agent Learning via Iterative Step-Level Process Refinement." ACL 2024.
>
> > **W3: I'm concerned that the author's hypotheses in lines 188-191 are untenable. It seems like saying that "the more confident the answer is for a given model, the more likely it is to be correct." and in this case the greedy answer should usually be selected?**
>
> We want to clarify that greedy autoregressive generation does not produce the most confident answer (i.e., a sequence) as scored by the LLM (maximum log P). Instead, it only generates the most confident token at the current generation step. As shown in Figure 2, around 60% of the samples yield more confident answers through non-myopic generation, illustrating that greedy answers are not always selected when aiming to find the most confident answer.
>
> Moreover, the hypothesis proposed in lines 188–191 suggests that: 1) LM next-token prediction is entirely myopic, and 2) LLMs, however, could internally perform non-myopic generation due to their extensive pre-training. We aim to discuss whether LLMs can perform non-myopic generation effectively; however, our analysis suggests otherwise.

---

> > ### Comment · Reviewer_iHG8 · 2024-11-23
> > **Thank you, I raised my score for 1 point.**
> >
> > Thank you for your feedback. The responses address most of my concerns. I will raise my score.

---

> > > ### Author Response · Authors · 2024-11-24
> > > **Thank you for your feedback**
> > >
> > > Thank you again for your valuable feedback to help us refine the quality of our paper!

---

### Official Review · Reviewer_fXex · 2024-11-06

**Soundness:** 2
**Presentation:** 2
**Contribution:** 2
**Rating:** 6
**Confidence:** 2

**Summary:**

This paper looks at LLM planning and reasoning from an optimal control perspective and proposes a novel method to enhance planning accuracy. The authors argue that LLMs face challenges in reliable and optimal planning due to their inherent myopic nature of autoregressive decoding. Predictive-Decoding leverages Model Predictive Control to re-weight LLM distributions based on foresight trajectories, aiming to mitigate early errors and promote non-myopic planning.

**Strengths:**

1. The idea of making the LLM to think long term (non-myopic) makes sense for LLM planing and reasoning
2. The author seems to perform different experimentation on different test-beds.

**Weaknesses:**

1. the presentation of this paper could be improved (for instance fig 1 is too many figures/text, could you make it simpler)?
2. the introduction of "Myopic Gap" seems to be interesting, but the section could be rewritten for better readability.
3. lacking larger llm model size for math and gsm experimentation.

**Questions:**

1. how does the number of FLOPS related to # of tokens? And also to the actual decoding/inference time?
2. how sensitive in the sampling temperature in determining the optimal solution parameter? Do you have to tune this for each test-beds?
3. it seems the diversity is still an issue even in the MPC problem right? how would you solve it?

---

> ### Author Response · Authors · 2024-11-21
> **Response to Reviewer fXex**
>
> Thank you for your careful review and constructive comments.
>
> We have revised our manuscript to incorporate your valuable suggestions, including:
> - Presentation of Figure 1 and Section 3
> - Additional results on larger LLM model size for math tasks.
>
> ----
> We address the concerns below:
> > **W1: the presentation of this paper could be improved (simplify Figure 1)?**
>
> Thanks for your suggestion! We have updated Figure 1 in the revised manuscript.
> > **W2: the introduction of "Myopic Gap" seems to be interesting, but the section could be rewritten for better readability.**
>
> We have rewritten Section 3 in the revised manuscript, providing more detailed explanations and a more comprehensive analysis of the results.
> > **W3: lacking larger llm model size for math and gsm experimentation.**
>
> Thanks for your suggestion. We add additional experiments for a larger LLM, Llama3-70B on MATH and GSM8K tasks. On both tasks, Predictive-Decoding surpasses myopic generation. MATH tasks show 5.4% improvement, and GSM8K shows 3% improvement, indicating a smaller yet significant improvement from non-myopic generation on stronger models. We have also added the results of Llama3-70B to Table 4 of the paper.
>
> | Model      | Method               | Inference FLOPS | Sample | GSM8K | MATH |
> |------------|----------------------|-----------------|--------|-------|------|
> | Llama3-70B | Autoregressive (PAL) | $53.7*10^{12}$      | N=1    | 90.1  | 43.8 |
> | Llama3-70B | Beam Search          | $343.7*10^{13}$     | N=1    | 91.4  | 48.1 |
> | Llama3-70B | Predictive-Decoding  | $115.5*10^{13}$     | N=1    | **93.1**  | **49.2** |
>
> > **Q1: how does the number of FLOPS related to # of tokens? And also to the actual decoding/inference time?**
>
> The number of FLOPS directly correlates with the average number of generated tokens during inference. The calculation of FLOPS is $6nP$, where $P$ is the number of parameters within the LLM, and $n$ refers to the number of tokens generated. This calculation process follows [1][2].
>
> [1] Kaplan, Jared, et al. "Scaling laws for neural language models." (2020).
>
> [2] Wu, Yangzhen, et al. "An empirical analysis of compute-optimal inference for problem-solving with language models." (2024).
>
> The actual decoding time is dependent on the inference architecture. For an inference framework without any acceleration, FLOPS reflects the actual decoding time. However, we use vLLM for implementation which supports batch-wise parallel decoding and prefix caching. The three algorithms are supported differently with vLLM acceleration:
> - Autoregressive: batch decoding (vLLM supported)
> - Beam Search: batch decoding + prefix caching (vLLM supported)
> - Predictive Decoding: step-level batch decoding (our implementation)
>
> Here is the calculated actual decoding time v.s FLOPS on HumanEval with Llama3-8B.
>
> | Configuration                  | Method              | Batch size | FLOPS  | Accelerated Decoding Time |
> |--------------------------------|---------------------|------------|--------|---------------------------|
> |  -                      | Autoregressive      | 164        | x 1    | x 1                       |
> | Beam size  = 8                 | Beam Search         | 164        | x 64   | x 9.1                     |
> | Sampling number=8              | Pass@8              | 164        | x 8    | x 2.2                     |
> | Foresight=6, Sampling number=8 | Predictive-Decoding | 164        | x 26.4 | x 16.1                    |
>
> We implement Predictive Decoding parallel inference such that all samples within the batch calculate the next step based on foresight, and then the entire batch proceeds to the next step together. While Predictive Decoding appears slower in the table, we emphasize that the runtime largely depends on the implementation. Currently, Predictive Decoding is not well-optimized because vLLM does not yet support prefix caching. We plan to adopt vLLM's implementation of beam search to further accelerate Predictive Decoding with prefix caching in the future.

---

> ### Author Response · Authors · 2024-11-21
> **Response to Reviewer fXex**
>
> > **Q2: how sensitive in the sampling temperature in determining the optimal solution parameter? Do you have to tune this for each test-beds?**
>
> Sampling temperature is important for performance. We have two sampling temperatures involved in our algorithm, the action selection temperature $\tau$, and the LLM generation temperature $\alpha$. Generally the algorithm is sensitive to both parameters, as we report below. However, we note that we do not tune them for each test-beds, we tune the parameters when using a **new LLM** or when a **new type of task** is present (math, coding, agent).
>
> The action selection temperature $\tau$ reflects the balance between local and global confidence. When $\tau=\infty$, the decoding results are exactly the same as autoregressive generation, while $\tau=0$ denotes that the exact maximum value of the global confidence is chosen. Most LLMs and reasoning tasks are very reliant on global confidence. Therefore the best $\tau$ is often close to 0.
>
> In practice, tuning of $\tau$ is dependent mainly on the model. This is due to different models are different in terms of myopia, e.g. Deepseek-coder is less myopic compared to Llama3. The optimal configuration for all models and tasks are shown in Appendix Table 10.
>
> $\alpha$ reflects basic LLM ability and sampling diversity. Normally, we use a balanced value for $\alpha$: between 0.6-1.0.
>
> To better illustrate sampling temperature sensitivity, we report results on 2 different tasks. A curve illustrating sensitivity is also available in Figure 5.
>
> | Configuration         | Model        | Method              | Tasks | Performances |
> |-----------------------|--------------|---------------------|-------|--------------|
> | $\alpha=0.6$            | Mistral-v0.3 | Autoregressive      | GSM8K | 53.4         |
> | $\alpha=0.6, \tau=0.1$  | Mistral-v0.3 | Predictive-Decoding | GSM8K | 59.7         |
> | $\alpha=0.6, \tau=0.01$ | Mistral-v0.3 | Predictive-Decoding | GSM8K | 64.5         |
> | $\alpha=1.0, \tau=0.05$ | Mistral-v0.3 | Predictive-Decoding | GSM8K | 64.5         |
> | $\alpha=1.0, \tau=0.01$ | Mistral-v0.3 | Predictive-Decoding | GSM8K | **66.7**         |
>
>
> | Configuration         | Model     | Method              | Tasks     | Performances |
> |-----------------------|-----------|---------------------|-----------|--------------|
> | $\alpha=0.6$            | Llama3-8B | Autoregressive      | HumanEval | 44.3         |
> | $\alpha=0.6, \tau=5.0$  | Llama3-8B | Predictive-Decoding | HumanEval | 49.1         |
> | $\alpha=0.6, \tau=1.0$  | Llama3-8B | Predictive-Decoding | HumanEval | 48.8         |
> | $\alpha=0.6, \tau=0.5$  | Llama3-8B | Predictive-Decoding | HumanEval | 48.2         |
> | $\alpha=0.6, \tau=0.1$  | Llama3-8B | Predictive-Decoding | HumanEval | 48.9         |
> | $\alpha=0.6, \tau=0.05$ | Llama3-8B | Predictive-Decoding | HumanEval | 53.8         |
> | $\alpha=0.6, \tau=0.01$ | Llama3-8B | Predictive-Decoding | HumanEval | **56.8**         |
> |$\alpha=0.1, \tau=0.05$ | Llama3-8B | Predictive-Decoding | HumanEval | 54.0         |
> | $\alpha=1.0, \tau=0.05$ | Llama3-8B | Predictive-Decoding | HumanEval | 44.1         |
>
> We can see for both tables, a too large $\tau$ results in significant performance drop and a small $\tau$ yields good results for both math and reasoning. However, the optimal configurations of $\alpha$ slightly vary across models.
>
> > **Q3: it seems the diversity is still an issue even in the MPC problem right? how would you solve it?**
>
> Generation diversity for reasoning tasks is a challenging issue for most algorithms. However, diversity is not undermined but improved with our method due to sampling twice. Please refer to lines 427-430 and Figure 5 for detailed analysis.

---

> > ### Author Response · Authors · 2024-11-26
> >
> > Dear Reviewer fXex,
> >
> > We would like to know if our response has addressed your concerns and questions. If you have any additional suggestions regarding the paper or our reply, please let us know. We’re happy to make further improvements.
> >
> > Thank you again for your time and effort!

---

### Author Response · Authors · 2024-11-21
**Updated Manuscript and Response to All Reviewers**

We sincerely thank all the reviewers for their feedback and constructive comments. We are pleased that the reviewers recognize the novelty and importance of non-myopic generation for LLM planning (R#1, R#2, R#3) as well as the solid/nice results across comprehensive tasks (R#1, R#2, R#3, R#4). In response to the reviewers’ comments, we have conducted three additional experiments (including one additional LLM, two search baselines, and an analysis) and revised the manuscript accordingly. **All changes are highlighted in blue in the updated PDF**. The updates are summarized as follows:

1. **Figure 1:** Revised the main figure and selected an example with less text. (R#1)
2. **Section 2:** Replaced the POMDP process with a decision process to avoid confusion. (R#2)
3. **Sections 3.1 & 3.2:** Added an explanation for the myopic gap formulation (R#1) and provided a more detailed analysis of the results.
4. **Section 4:** Included additional details about the evaluation process. (R#4)
5. **Section 5.2, Table 4:** Added results for Llama-3-70b on GSM8K and Math. (R#1)
6. **Section 5.3, Table 8:** Included results for A* search and Tree of Thought. (R#4)
7. **Appendix C:** Added an analysis of self-reflection abilities. (R#4)
8. **Appendix E.1:** Conducted a case study to illustrate how myopic generation leads to planning failure. (R#2)
9. **Appendix E.2:** Added an analysis of failure cases. (R#3)

We have addressed the queries and suggestions raised by each reviewer in detail and provided clarifications where necessary. We greatly appreciate the reviewers’ valuable contributions, which have helped us improve the quality of our manuscript. Should further clarification be required to assist in advancing our score, please do not hesitate to let us know.

Thank you once again for your thoughtful review and feedback!

---

### Author Response · Authors · 2024-11-24
**A Kind Reminder for Reading the Response**

Thank you again for your comments! We have added many revisions to address your concerns. Since the rebuttal period is closing very soon, can you please check the response to see whether it mitigates your conerns? We would greatly appreciate that!

Thank you,

The Authors

---

### Meta-Review · Area_Chair_nkkG · 2024-12-23

**Metareview:**

LLMs face challenges in ensuring optimal planning due to their inherent myopic nature of autoregressive decoding. Motivated by these challenges, the authors propose a novel method Predictive-Decoding inspired by model predictive control (MPC) to mitigate early errors and enhance planning accuracy.

Although the presentation of the original version is unclear to some extent and lacks comparisons with current search methods, this paper presents an intuitive integration of MPC with LLM. The authors offer an interesting method supported by experimental results, including failure cases, and the authors' response addresses most of the reviewers’ questions.

**Additional Comments On Reviewer Discussion:**

See the Official Comment *Updated Manuscript and Response to All Reviewers* by Authors

---

### Decision · Program_Chairs · 2025-01-22

Accept (Poster)